# reAR: Rethinking Visual Autoregressive Models via Generator-Tokenizer Consistency Regularization

**Qiyuan He**[1], **Yicong Li**[1*], **Haotian Ye**[2], **Jinghao Wang**[3], **Xinyao Liao**[1]
**Pheng-Ann Heng**[3], **Stefano Ermon**[2], **James Zou**[2], **Angela Yao**[1*]
[1]National University of Singapore    [2]Stanford University
[3]The Chinese University of Hong Kong

## ABSTRACT

Visual autoregressive (AR) generation offers a promising path toward unifying vision and language models, yet its performance remains suboptimal against diffusion models. Prior work often attributes this gap to tokenizer limitations and rasterization ordering. In this work, we identify a core bottleneck from the perspective of generator-tokenizer inconsistency, i.e., the AR-generated tokens may not be well-decoded by the tokenizer. To address this, we propose reAR, a simple training strategy introducing a token-wise regularization objective: when predicting the next token, the causal transformer is also trained to recover the visual embedding of the current token and predict the embedding of the target token under a noisy context. It requires no changes to the tokenizer, generation order, inference pipeline, or external models. Despite its simplicity, reAR substantially improves performance. On ImageNet, it reduces gFID from 3.02 to 1.86 and improves IS to 316.9 using a standard rasterization-based tokenizer. When applied to advanced tokenizers, it achieves a gFID of 1.42 with only 177M parameters, matching the performance with larger state-of-the-art diffusion models (675M).

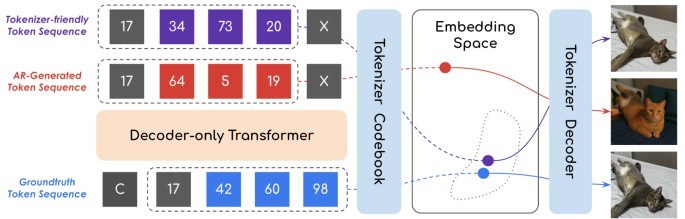
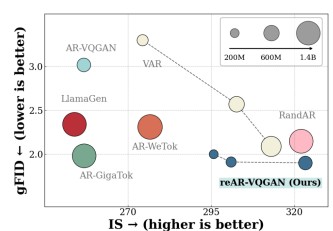

(a) Visual autoregressive generation suffers from generator–tokenizer inconsistency: (1) Due to **exposure bias**, the AR model is more likely to generate token sequences unseen by the tokenizer; (2) Being **embedding unaware**, the embedding sequence of the generated discrete tokens may also be unfamiliar to the tokenizer, resulting in a cat in an unnatural pose, with its lower body flipped and the belly facing upward. The top and bottom images can still appear similar despite differing token indices, since distinct token sequences may map to nearby embeddings.

(b) With *generator-tokenizer consistency regularization*, reAR with fewer parameters significantly improves over vanilla AR (gFID: 3.02 to 1.86, IS: 256.2 to 316.9) and even surpasses methods based on advanced tokenization and sophisticated generative paradigm.

Figure 1: **Generator-tokenizer inconsistency is the bottleneck in the visual autoregressive model.**

## 1 INTRODUCTION

Autoregressive (AR) models, using a decoder-only transformer with the objective of next token prediction, are state-of-the-art for natural language generation (Team et al., 2023; Achiam et al., 2023). For image generation, however, AR models are less competitive than diffusion models (Dhariwal

---

*Corresponding Authors.

& Nichol, 2021; Peebles & Xie, 2023; Ma et al., 2024; Yu et al., 2024c). There is great interest in advancing visual autoregressive models to unify the language and visual modalities into a single generative framework (Bai et al., 2024; Team, 2024; Chung et al., 2024).

Scrutinizing the current design in visual AR, the dominant paradigm is to convert images into discrete tokens and train an autoregressive model on the converted token sequences. Specifically, a tokenizer is trained to split an image (or the feature) into patches and utilizes them into a sequence of discrete tokens (Esser et al., 2021; Sun et al., 2024; Luo et al., 2024), which it can use to reconstruct the original image. A decoder-only transformer using a causal mask is then trained on this token sequence in raster-scan order with the objective of next-token prediction. Unfortunately, this paradigm typically results in suboptimal performance compared to the diffusion model (Dhariwal & Nichol, 2021; Peebles & Xie, 2023; Ma et al., 2024; Yu et al., 2024c). Previous works have analyzed the performance gap from the perspective of tokenization, including token decomposition (Tian et al., 2024; Yu et al., 2024b; Bachmann et al., 2025; Pan et al., 2025) and sequence order (Pang et al., 2025; Yu et al., 2024a), rather than the whole system of visual autoregressive generation.

In this work, we provide a unified perspective on the key bottleneck of visual AR through the lens of **generator-tokenizer inconsistency**, which refers to the challenge that the autoregressive model might generate a token sequence that is hard for the tokenizer to decode back to an image. Specifically, we examine two sources of the inconsistencies inherited from the generated token sequence.

Firstly, the generated token sequence can be *unseen* by the tokenizer due to **exposure bias**. In autoregressive training, each token is predicted given the ground-truth context (teacher forcing), but at inference, the context consists of the model's own predictions. Early mistakes then compound and lead to sequences never observed during training. While exposure bias is well studied in language (Bengio et al., 2015; Wang & Sennrich, 2020), it is *amplified* in visual AR. Text tokens are themselves the final output, so even an unseen sequence may still be semantically coherent. By contrast, visual tokens are decoded into images: a single wrong token can corrupt future predictions and decode into a token sequence never seen by the tokenizer during training, spreading structural artifacts across the image. As shown in Figure 1(a), an early misprediction (e.g., 42'→64') cascades through subsequent tokens and yields a cat in an unnatural pose with a different coat color.

Secondly, the AR model suffers from **embedding unawareness**. During training, it optimizes only the discrete token indices without considering how these tokens are embedded by the tokenizer. However, the decoded image quality depends on the embeddings of the generated tokens rather than their indices alone, as shown in Figure 1(a). This unawareness leads to two issues: (i) even if two tokens are close in the embedding space, the model can only infer this relation indirectly from co-occurrence statistics, which is data-inefficient. and (ii) the embedding of an incorrect token is unconstrained by the ground-truth embedding, which can cause the overall sequence embedding to drift far from the training distribution of the tokenizer decoder. As illustrated in Figure 1(a), although the purple and red sequences contain the same number of incorrect tokens, the one with embeddings closer to the ground truth generates a decoded image of higher quality.

In this regard, we propose reAR, a unified training framework that explicitly regularizes the model toward tokenizer-friendly behavior. Concretely, we introduce two complementary strategies: 1) **Noisy Context Regularization** that exposes the model to perturbed context during training, reducing its reliance on clean contexts and improving robustness to imperfect histories at test time, thereby alleviating the model's tendency to generate unseen token sequence; 2) **Codebook Embedding Regularization** that aligns the generator's hidden states with the tokenizer's embedding space, which encourages the generator to be aware of how tokens are decoded into visual patches. By learning to predict the embeddings rather than only discrete indices, even if the generator generates an unseen token sequence, the corresponding embedding sequence is optimized to be more compatible with the tokenizer. Combining them together, the *token-wise consistency regularization* can guide visual AR to be friendly to the tokenizer by predicting the visual embedding in a robust manner.

Building on reAR, we conduct extensive experiments comparing it against other generative frameworks. To show that reAR generalizes beyond specific tokenizers, we apply it to non-standard designs such as TiTok (Yu et al., 2024b) and AliTok (Wu et al., 2025). When combined with standard rasterization-order AR, reAR outperforms prior autoregressive methods even when those rely on sophisticated tokenizers, as Figure 1 (b) shows. Under the same model size and training budget, it also surpasses alternative paradigms such as MAR (Li et al., 2024), VAR (Tian et al., 2024), and

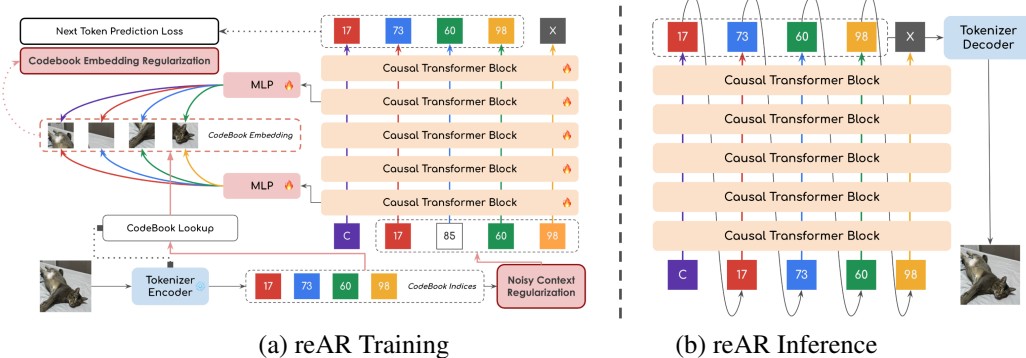

(a) reAR Training
(b) reAR Inference

Figure 2: **Overview of reAR, a plug-and-play framework that is agnostic to the visual tokenizer.**

SiT (Ma et al., 2024). Furthermore, when paired with a tokenizer tailored for causal AR modeling (Wu et al., 2025), reAR achieves FID = 1.42 with only 177M parameters—competitive with the diffusion model REPA, which requires external representations and 675M parameters.

Our contributions can be summarized as follows:

- We identify the inconsistency between generator and tokenizer, where tokenizer fails to decode the generated token sequence, as the bottleneck of visual autoregressive generation;

- We propose reAR, a plug-and-play training regularization that introduces visual inductive bias from the tokenizer and alleviates exposure bias to train the visual autoregressive model;

- We demonstrate that reAR significantly improves visual autoregressive generation across different tokenizers (e.g., on VQGAN, FID improves from 3.02 to 1.86) and even surpasses more sophisticated generative models, using far fewer parameters.

## 2 PRELIMINARIES

Visual autoregressive generation is commonly divided into two components: (1) A visual tokenizer to tokenize the image; (2) An autoregressive model to sample the token sequence.

**Visual Tokenizer**. Visual tokenizers compress image pixels into discrete token sequences. The most commonly adopted methods are patch-based tokenizers (Esser et al., 2021; Sun et al., 2024; Yu et al., 2023; Chang et al., 2022). The tokenizer includes three parts: Encoder $\mathcal{E}$, Quantizer $\mathcal{Q}$ and Decoder $\mathcal{D}$. Formally, a given image $\mathbf{I} \in \mathbb{R}^{3 \times H \times W}$ is converted to a feature $\hat{\mathbf{z}} \in \mathbb{R}^{c \times h \times w}$ with the encoder $\mathcal{E}$ where $h < H, w < W$. It's then processed into quantized embedding $\mathbf{z^q} \in \mathbb{R}^{c \times h \times w}$ via the quantizer $\mathcal{Q}$ and decoded back to reconstruct image $\hat{\mathbf{I}}$ by the decoder $\mathcal{D}$:

$$\hat{\mathbf{z}} = \mathcal{E}(\mathbf{I}), \quad \mathbf{z^q} = \mathcal{Q}(\hat{\mathbf{z}}), \quad \hat{\mathbf{I}} = \mathcal{D}(\mathbf{z^q}) \tag{1}$$

The vector quantization is performed *element-wise* with a codebook $\mathcal{Z} = \{\mathbf{z}_1, \mathbf{z}_2, \ldots, \mathbf{z}_K\} \subset \mathbb{R}^{c \times h \times w}$ by looking up the closest entry. Formally:

$$\mathbf{z^q}_{ij} = \arg\min_{\mathbf{z}_k \in \mathcal{Z}} \left\| \hat{\mathbf{z}}_{ij} - \mathbf{z}_k \right\|, \quad \mathbf{x}_{ij} = \arg\min_{k \in \{1,\ldots,K\}} \left\| \hat{\mathbf{z}}_{ij} - \mathbf{z}_k \right\|. \quad i = 1, \ldots, h, \quad j = 1, \ldots, w. \tag{2}$$

where $\mathbf{x}_{ij}$ forms the discrete token (indices such as 17 and 73). In the standard approach, it's arranged into 1D token sequence via row-major rasterization order, i.e., $\{\mathbf{x}_{11}, \ldots, \mathbf{x}_{1w}, \mathbf{x}_{21}, \ldots, \mathbf{x}_{h1}, \ldots, \mathbf{x}_{hw}\}$. The autoregressive model can then be trained on it..

**Autoregressive Model**. To model the distribution of a sequence of signal $\mathbf{x}_{1:N} = \{\mathbf{x}_1, \mathbf{x}_2, \ldots, \mathbf{x}_N\}$, the autoregressive model $p_\theta$ aims to maximize the likelihood of the next token under teacher forcing:

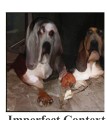 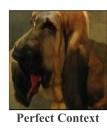 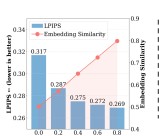 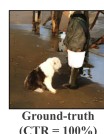 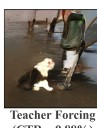 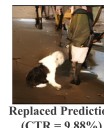

| | |
|---|---|
| (a) Tokenizer is sensitive to the error of generated tokens from exposure bias | (b) Tokenizer is sensitive to the embedding of generated tokens |

Figure 3: **Token sequence with the same correct token ratio** (CTR) **under teacher forcing can be decoded into images with different quality.** Under the same CTR, (a) The images decoded from imperfect context is much less similar to the ground truth than the one from perfect context; (b) Replacing incorrect token with other incorrect tokens but with more similar embedding of the correct token, the generated image can be more similar to ground truth than original prediction.

$$\theta = \arg\max_{\theta} \log p_\theta(\mathbf{x}_{1:N}) = \arg\max_{\theta} \sum_{i=1}^{N} \log p_\theta(\mathbf{x}_i | \mathbf{x}_1, \mathbf{x}_2, \ldots, \mathbf{x}_{i-1}) \tag{3}$$

During inference, the model then decodes the sequence one by one. The $t$th token is sampled from the context $\mathbf{x}_{1:t-1}$ by $\mathbf{x}_t \sim p(\cdot \mid \mathbf{x}_{1:t-1})$ under free running. In visual autoregressive generation, after sampling a sequence $\hat{\mathbf{x}}$ from $p_\theta$, it's decoded into $\hat{I}$ as the final generated image by the tokenizer decoder $\mathcal{D}$.

## 3 reAR: Regularizing Consistency in Visual AR

Different from natural language, $\hat{\mathbf{x}}$ is not the final generated result in visual autoregressive generation. Therefore, inconsistency between the generator and decoder can lead to unsatisfying results even if the autoregressive model is trained well. For example, when sampling an unseen or rare sequence $\hat{\mathbf{x}}$ in the training dataset of the tokenizer, it's possible that the sequence $\hat{\mathbf{x}}$ cannot be properly decoded by decoder $\mathcal{D}$ and affect the final generated results. We hypothesize that the inconsistency between the tokenizer and generator is the main obstacle to performance. A promising solution is to train the AR model such that it can generate a token sequence that is friendly to the tokenizer.

To verify our hypothesis, we investigate and quantitatively analyze how the existing visual autoregressive model suffers from the inconsistency in Section 3.1. Based on the observations, we propose **reAR: regularizing token-wise consistency of visual AutoRegressive generation**, a plug-and-play regularized training method designed for a visual autoregressive model. In summary, reAR introduces visual embedding looked up from a discrete tokenizer to the hidden feature of the generator under a noisy context. Despite its simplicity, reAR allows the autoregressive model to leverage visual signals that are compatible with the tokenizer and reduce inconsistent behavior significantly.

### 3.1 Understanding the Bottleneck of Visual Autoregressive Generation

The performance of an autoregressive model can be assessed through the quality of generated tokens $\hat{x}_{1:n}$ with the ground-truth sequence $x_{1:n}$ by the **correct token ratio** (CTR), where $\text{CTR}(\hat{x}_{1:n}, x_{1:n}) = \frac{1}{n}\sum_{i=1}^{n} \mathbf{1}\{\hat{x}_i = x_i\}$. While CTR is widely used to indicate the performance, the token sequence is only an intermediate representation in visual autoregressive generation, and the final output is actually the decoded image. To evaluate end-to-end quality, we instead measure LPIPS (Zhang et al., 2018) between the images decoded from two token sequences. We consider that the inconsistencies between training and inference can be observed from inconsistencies between CTR and LPIPS. In the following, two controlled experiments demonstrate that generated token sequences with similar CTR can result in images of different quality. This inconsistency is also reflected by other metrics for the AR model, such as **perplexity**, with details in the Appendix B.

**Amplified exposure bias.** Exposure bias is a well-known issue in sequence models (Bengio et al., 2015; Wang & Sennrich, 2020): during training with teacher forcing, the model predicts the next token given the *ground-truth* context, whereas at inference it must condition on its *own* predictions,

which may contain errors. In visual autoregressive generation, we hypothesize that the visual tokenizer amplifies this effect since exposure bias leads to more unseen token sequences and spreads structural error in the pixel space. To verify it, consider a token sequence $x_{1:n}$ decoded from an image with a ground-truth token ratio $r \in [0, 1]$. We compare two decoding protocols: (1) *Perfect context (front-loaded).* Fix the first $\lfloor rn \rfloor$ tokens to ground truth, i.e., $x_{1:\lfloor rn \rfloor}$, and let the AR model generate the remainder. This minimizes exposure bias for a given $r$, since the context remains clean until step $\lfloor rn \rfloor$. (2) *Imperfect context (uniformly interleaved).* Sample a mask $M \subseteq \{1, \ldots, n\}$ with $|M| = \lfloor rn \rfloor$ uniformly at random. During decoding at the $t$th step, it uses ground truth token $x_t$ if $t \in M$, otherwise samples the token from the AR model. This introduces earlier contamination of the context, thereby increasing exposure bias compared to *Perfect context* with similar CTR.

Since both protocols fix the number of ground-truth tokens at $\lfloor rn \rfloor$, any difference in downstream quality reflects sensitivity to exposure bias rather than token-level accuracy. Results are shown in Figure 3 (a). For comparable CTR, imperfect context consistently yields higher LPIPS than perfect context. Qualitatively, an imperfect context leads to images that deviate significantly from the original, whereas a perfect context yields better prediction, i.e., the layout of the dog is more similar. This highlights that alleviating exposure bias is essential in visual autoregressive generation.

**Embedding unawareness.** During training, the AR model is optimized only for token correctness, whereas the tokenizer decoder operates in embedding space. We hypothesize that even if a predicted token is incorrect, if its embedding is close to that of the correct token, the decoded image may still retain high visual quality. To verify this, we introduce a replacement ratio $r'$. Given a ground-truth sequence $x_{1:n}$, the AR model predicts $\hat{x}_{1:n}$ with teacher forcing. For each incorrect prediction ($\hat{x}_i \neq x_i$), we replace $\hat{x}_i$ with probability $r'$ by another incorrect token $x'_i \neq x_i$ whose embedding $z^{q'}_i$ is closest to the correct embedding $z^q_i$ under cosine similarity $d(\cdot, \cdot)$, i.e., $z^{q'}_i = \arg\min_{z^q \in \mathcal{Z} \setminus \{z^q_i\}} d(z^q, z^q_i)$. This replacement leaves CTR unchanged.

Figure 3(b) presents the results. As $r'$ increases, the average embedding similarity improves and LPIPS decreases markedly. Qualitatively, as shown on the right of Figure 3(b), such replacements without altering CTR can yield decoded images more faithful to the ground truth (e.g., clearer prediction of shirts and human legs). This suggests that incorporating tokenizer embeddings into the training of the AR model could potentially improve consistency between them.

A straightforward approach to increase generator-tokenizer inconsistency is to reuse the tokenizer's codebook embeddings in the embedding layer or prediction head of the AR model. However, this method commonly results in suboptimal performance without sophisticated design of the tokenizer (Weber et al., 2024; Yu et al., 2023). We hypothesize that such a rigid integration is not ideal: it may constrain the scalability of a large AR model with a smaller tokenizer, and the codebook embeddings themselves may not be the optimal representations for the primary task of next-token prediction. It's required to introduce the embedding into the model in a less constrained manner.

## 3.2 GENERATOR-TOKENIZER CONSISTENCY REGULARIZATION

These findings reveal training–inference inconsistencies: maximizing correctness to predict token indices alone is insufficient for visual AR models. Proper inductive bias is required to train the generator such that the generated token sequence is more consistent with the tokenizer during inference. Meanwhile, injecting this inductive bias should remain lightweight to ensure good cross-architecture generalization and full compatibility with existing AR training and inference pipelines.

To address these challenges, reAR introduces token-wise consistency regularization during training of the visual AR model. Specifically, the decoder-only transformer is trained to perform next-token prediction under noisy contexts, while its hidden representations are regularized by the visual embeddings of the correct current token at a shallow layer and the correct next token at a deep layer. This encourages the AR model to interpret current tokens similar to the tokenizer while improving robustness to exposure bias, then predicting the next token embedding compatible with the decoder. Below we denote the AR model as $p_\theta$, the tokenizer codebook as $\mathcal{Z} = \{\mathbf{z}_1, \mathbf{z}_2, \ldots, \mathbf{z}_K\}$, the training dataset as $\mathcal{X}_{\text{train}}$, and the discrete token sequence as $\mathbf{x} = \{\mathbf{x}_1, \mathbf{x}_2, \ldots, \mathbf{x}_N\}$.

**Noisy Context Regularization.** While techniques such as scheduled sampling (Bengio et al., 2015) can mitigate exposure bias, we choose a simple approach that preserves parallel training of the transformer. Specifically, we apply uniform noise to the input, denoted by $q_\epsilon(\tilde{\mathbf{x}} \mid \mathbf{x})$. Formally:

$$\tilde{\mathbf{x}}_i = (1 - b_i)\,\mathbf{x}_i \;+\; b_i\,\mathbf{u}_i, \quad b_i \sim \text{Bernoulli}(\epsilon), \quad \mathbf{u}_i \sim \text{Uniform}\big(\{1, \ldots, K\}\big). \tag{4}$$

where $b_i$ is a Bernoulli random variable with probability $\epsilon$, and $\mathbf{u}_i$ is sampled uniformly from the codebook indices. In practice, the choice of $\epsilon$ strongly affects training stability. To ensure the AR model is exposed to sequences with varying noise levels, we sample $\epsilon \sim U(0, f(t))$ for each token sequence, where $t \in [0, 1]$ denotes the normalized training progress. Here, $f : [0, 1] \to [0, 1]$ is an annealing schedule that controls the maximum noise level over training. The AR model is then trained to predict the next correct token based on the noisy context. Formally:

$$\mathcal{L}'_{\text{AR}}(\theta) = -\mathbb{E}_{\mathbf{x} \in \mathcal{X}_{\text{train}}, \tilde{\mathbf{x}} \sim q_\epsilon(\cdot|\mathbf{x}), \epsilon \sim U(0, f(t))} \sum_{i=1}^{N} \log p_\theta(\mathbf{x}_i | \tilde{\mathbf{x}}_{i-1}, \ldots, \tilde{\mathbf{x}}_1) \tag{5}$$

Empirically, we found that the annealing uniform noisy augmentation can stabilize training compared to noisy augmentation with a fixed ratio. We provide detailed ablation in Section 4.3.

**Codebook Embedding Regularization**. Instead of directly applying codebook embedding, we propose to add a regularization task as **recover current embedding** and **predict next embedding**. Specifically, we apply a trainable MLP layer $h_\phi$ to project the hidden feature into the target space in the same dimension of visual embedding. For the simplicity of notation, we use $\mathbf{w}_\theta^l(\tilde{\mathbf{x}})$ to represent the feature at the shallow layer $l$ and $\mathbf{w}_\theta^{l'}(\tilde{\mathbf{x}})$ as the one at the deep layer $l'$. To be aligned with the design of decoder-only transformer, the objective of the shallow layer $\mathbf{w}_\theta^l(\tilde{\mathbf{x}})$ is to predict the embedding of current token and $\mathbf{w}_\theta^{l'}(\tilde{\mathbf{x}})$ is to predict the next token. Formally:

$$\mathcal{L}_{\text{re}}(\theta, \phi; t) = \mathbb{E}_{\substack{\mathbf{x} \sim \mathcal{X}_{\text{train}}, \\ \tilde{\mathbf{x}} \sim q_\epsilon(\cdot|\mathbf{x}), \\ \epsilon \sim U(0, f(t))}} \sum_{i=1}^{N-1} \Big[ d\big(h_\phi^i(\mathbf{w}_\theta^l(\tilde{\mathbf{x}})), z_{\mathbf{x}_i}\big) \;+\; d\big(h_\phi^i(\mathbf{w}_\theta^{l'}(\tilde{\mathbf{x}})), z_{\mathbf{x}_{i+1}}\big) \Big]. \tag{6}$$

where $d(\cdot, \cdot)$ is cosine distance to evaluate the distance between different features, $h_\phi^i$ refers the mapping from the feature of the $i^{th}$ current token to the embedding space, $z_{\mathbf{x}_i}$ is the embedding of current token and $z_{\mathbf{x}_{i+1}}$ is the embedding of the next token looked up from the codebook. In the implementation, we apply the regularization on the layers that are originally most closely to the embedding of the tokenizer in the vanilla AR (i.e, the 1st layer for encoding regularization and the 15th layer for decoding regularization) to avoid potential conflicts on the primary task of next-token prediction. Intuitively, we place the encoding regularization at the first layer to ensure no downstream information required for predicting the next token is suppressed or overwritten, and apply the decoding regularization in a deep but not final layer, since the raw tokenizer embedding is not necessarily the best latent representation for prediction. By default, we regularize at three-quarters of the model depth, which works well across architectures though the exact layer for decoding regularization is flexible. We provide more analysis on the codebook embedding regularization in Section 4.3 and Appendix C.2.

**Generator-Tokenizer Consistency Regularization**. Combing Noisy Context Regularization and Codebook Embedding Regularization, the object of reAR is:

$$\mathcal{L}_{\text{reAR}}(\theta, \phi; t) = \mathcal{L}'_{\text{AR}}(\theta; t) + \lambda \mathcal{L}_{\text{re}}(\theta, \phi; t), \tag{7}$$

where $\lambda$ is the weight of the regularization term. Notice that we align the hidden feature of noisy tokens to the embedding of the ground truth token as well, which further encourages the autoregressive model to predict codebook embedding in a robust manner. **This joint effect** is important to boost the performance of visual autoregressive generation. We provide detailed ablation in Section 4.3.

Table 1: **Results on 256×256 class-conditional generation on ImageNet-1K.** "Mask." indicates masked generation; "Tok." denotes non-standard tokenization; "Rand." denotes randomized order; "Raster." denotes rasterization order. "†" indicates that the model is not provided and it's trained with our implementation. $\text{BPP}_{16} = 16 \times \text{BPP}$ (bits per pixel) measures the compression rate of discrete tokenizers and is not applicable ("N/A") to continuous tokenizers. "#Params" is the number of model parameters. "↑" and "↓" indicate whether higher or lower values are better, respectively.

| Training Paradigm | Generation Model | Tokenizer Type | Tokenizer $\text{BPP}_{16}$ ↓ | Training Epochs | #Params.↓ | FID↓ | IS↑ |
|---|---|---|---|---|---|---|---|
| Diffusion | LDM-4 (Rombach et al., 2022) | Patch-VAE | N/A | 200 | 400M | 3.60 | 247.7 |
| | DiT-XL (Peebles & Xie, 2023) | Patch-VAE | N/A | 1400 | 675M | 2.27 | 278.2 |
| | SiT-XL (Ma et al., 2024) | Patch-VAE | N/A | 800 | 675M | 2.06 | 270.3 |
| | REPA (Yu et al., 2024c) | Patch-VAE | N/A | 800 | 675M | 1.42 | 305.7 |
| MAR | MAR-L (Li et al., 2024) | Patch-VAE | N/A | 800 | 479M | 1.98 | 290.3 |
| | MAR-H (Li et al., 2024) | Patch-VAE | N/A | 800 | 943M | 1.55 | 303.7 |
| Mask. | MaskGIT-re Chang et al. (2022) | Patch-VQ | 0.625 | 300 | 227M | 4.02 | 355.6 |
| | MAGVIT-v2 (Yu et al., 2023) | Patch-VQ | 1.125 | 1080 | 307M | 1.78 | 319.4 |
| | Maskbit (Weber et al., 2024) | Patch-LFQ | 0.875 | 1080 | 305M | 1.52 | 328.6 |
| | Mask-TiTok-64 (Yu et al., 2024b) | TiTok | 0.188 | 800 | 177M | 2.48 | 214.7 |
| | Mask-TiTok-128 (Yu et al., 2024b) | TiTok | 0.375 | 800 | 287M | 1.97 | 281.8 |
| VAR | VAR-d20 (Tian et al., 2024) | VAR | 1.992 | 350 | 600M | 2.57 | 302.6 |
| | VAR-d30 (Tian et al., 2024) | VAR | 1.992 | 350 | 2.0B | 1.92 | 323.1 |
| Rand. Causal AR | RAR-B (Yu et al., 2024a) | Patch-VQ | 0.625 | 400 | 261M | 1.95 | 290.5 |
| | RAR-L (Yu et al., 2024a) | Patch-VQ | 0.625 | 400 | 461M | 1.70 | 299.5 |
| | RAR-XL (Yu et al., 2024a) | Patch-VQ | 0.625 | 400 | 955M | 1.50 | 306.9 |
| | RandAR-L (Pang et al., 2025) | Patch-VQ | 0.875 | 300 | 343M | 2.55 | 288.8 |
| | RandAR-XL (Pang et al., 2025) | Patch-VQ | 0.875 | 300 | 775M | 2.25 | 317.8 |
| | RandAR-XXL (Pang et al., 2025) | Patch-VQ | 0.875 | 300 | 1.4B | 2.15 | 322.0 |
| Tok. Causal AR | AR-FlexTok-XL (Bachmann et al., 2025) | FlexTok | 0.125 | 300 | 1.3B | 2.02 | – |
| | AR-GigaTok-XXL (Xiong et al., 2025) | GigaTok | 0.875 | 300 | 1.4B | 1.98 | 256.8 |
| | AR-WeTok-XL (Zhuang et al., 2025) | WeTok | 1.667 | 300 | 1.5B | 2.31 | 276.6 |
| Raster. Causal AR | VQGAN-re (Esser et al., 2021) | Patch-VQ | 0.875 | 100 | 1.4B | 5.20 | 280.3 |
| | Open-MAGVIT-v2 (Luo et al., 2024) | Patch-LFQ | 1.125 | 300 | 1.5B | 2.33 | 271.8 |
| | LlamaGen-XL (Sun et al., 2024) | Patch-VQ | 0.875 | 300 | 775M | 2.62 | 244.1 |
| | LlamaGen-XXL (Sun et al., 2024) | Patch-VQ | 0.875 | 300 | 1.4B | 2.34 | 253.9 |
| | AR-L† (Yu et al., 2024a) | Patch-VQ | 0.625 | 400 | 461M | 3.02 | 256.2 |
| | reAR-S | Patch-VQ | 0.625 | 400 | 201M | 2.00 | 295.7 |
| | reAR-B | Patch-VQ | 0.625 | 400 | 261M | 1.91 | 300.9 |
| | reAR-L (cfg=10.0/11.0) | Patch-VQ | 0.625 | 400 | 461M | 1.86/1.90 | 316.9/323.2 |

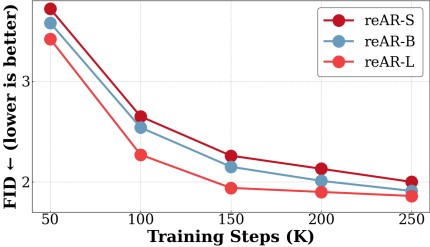

Figure 4: **Scaling Effect of reAR.** As model size increases, the FID at each training step decreases consistently.

Table 2: **Superior generalization ability**. reAR adapts to different tokenizers and achieves state-of-the-art performance with smaller models.

| Model | Epochs | Params. | FID ↓ |
|---|---|---|---|
| Maskbit (Weber et al., 2024) | 1080 | 305M | 1.52 |
| REPA (Yu et al., 2024c) | 800 | 675M | 1.42 |
| AR-TiTok-b64 (Yu et al., 2024b) | 400 | 261M | 4.45 |
| RAR-TiTok-b64 (Yu et al., 2024a) | 400 | 261M | 4.07 |
| reAR-TiTok-b64 | 400 | 261M | 4.01 |
| AR-AliTok-B (Wu et al., 2025) | 800 | 177M | 1.50 |
| RAR-B-AliTok (Yu et al., 2024a) | 800 | **177M** | 1.52 |
| reAR-B-AliTok | 800 | **177M** | **1.42** |

## 4 EXPERIMENTS & ANALYSIS

### 4.1 EXPERIMENTAL SETUP

Below we provide a brief of our experimental setup, and more details are in Appendix A.

**Dataset and evaluation.** We evaluate reAR on ImageNet-1K at $256\times256$ using the ADM protocol (Dhariwal & Nichol, 2021). Each model generates 50k images with classifier-free guidance (Ho & Salimans, 2022). We report FID (lower is better) (Heusel et al., 2017) and IS (higher is better) (Salimans et al., 2016), and compare training efficiency by epochs and parameters needed to reach the same quality. Baselines span diffusion, masked generation (continuous and discrete), VAR, randomized-order AR, advanced-tokenizer AR, and standard raster AR (see Table 1).

**Model configuration.** We use MaskGIT VQGAN (Chang et al., 2022) (rFID= 1.97) as a tokenizer and a DiT-style (Peebles & Xie, 2023) AR backbone. We report reAR-S/B/L with 20/24/24 causal Transformer layers and hidden sizes 768/768/1024. To evaluate the generalization of reAR, we also

pair it with TiTok (Yu et al., 2024b) and with AliTok (Wu et al., 2025) using their original setting. Additionally, we also verify the effectiveness of our method on non-standard causal AR model such as VAR (Tian et al., 2024) with more details in the Appendix A.

**Training.** All models are trained for 400 epochs on 8 A800 GPUs (batch size 2048) with AdamW (Loshchilov & Hutter, 2017), gradient clipping (norm= 1), and accumulation. The learning rate warms to $4 \times 10^{-4}$ over the first 100 epochs, then decays to $1 \times 10^{-5}$ for the remaining 300 epochs. Class labels are dropped with probability 0.1 to enable classifier-free guidance at inference.

**reAR implementation.** We apply a linear schedule for annealing noise augmentation. Embedding regularization is implemented using a 2-layer MLP (hidden size 2048, weight $\lambda$=1): the shallow layer regularizes the current embeddings at $l$=0, while the deeper layer regularizes the decoding features at $\frac{3}{4}$ depth of the whole transformer ($l' = 15/18/18$ for reAR-S/B/L).

## 4.2 MAIN RESULTS

**Generation Quality.** Table 1 shows that reAR achieves strong results even with a standard raster-order AR model and a simple 2D patch tokenizer. reAR-S outperforms prior raster AR models like LlamaGen-XL (Sun et al., 2024) (FID 2.00 vs. 2.34; IS 295.7 vs. 253.9) using only 14% of the parameters (201M vs. 1.4B), and surpasses advanced-tokenizer AR models such as WeTok (Zhuang et al., 2025) with just 13–15% of their size. It matches RAR (Yu et al., 2024a) and outperforms RandAR (Pang et al., 2025) under similar scales, and reAR-L exceeds MAR-L and VAR-d30 (Li et al., 2024; Tian et al., 2024). While diffusion and masked-generation models remain strong, reAR narrows the gap with far fewer training epochs. More qualitative results are shown in Appendix F.

**Generalization.** We also evaluate reAR on non-standard tokenizers TiTok (Yu et al., 2024b) and AliTok (Wu et al., 2025). Unlike RAR (Yu et al., 2024a), which helps mainly on bidirectional tokenization, reAR consistently improves performance on both bidirectional (TiTok: $4.45 \rightarrow 4.01$) and unidirectional (AliTok: $1.50 \rightarrow 1.42$) tokenizers. Notably, it approaches diffusion-based REPA (Yu et al., 2024c) and outperforms Maskbit while using far fewer parameters (177M vs. 675M/305M).

**Scaling Effect.** We also study if the scaling behavior of the original AR model maintains with reAR. Specifically, we plot the FID under different training epochs for each model size. As Figure 4 shows, the FID consistently decreases as model size and training iteration increase, revealing the potential of reAR on large-scale visual AR models.

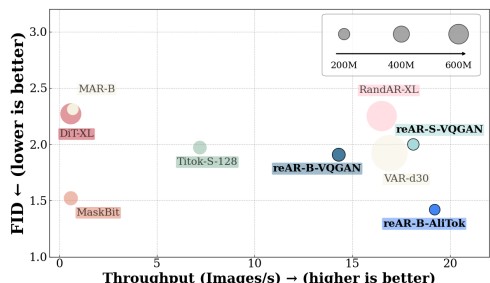

Figure 5: **Sampling Speed.** Comparison of different methods on FID and throughput (images/sec).

**Sampling Speed.** Like other autoregressive models (Sun et al., 2024; Luo et al., 2024), reAR benefits from KV-cache to achieve high sampling speed. We measure throughput on a single A800 GPU with batch size 128 (Figure 5). With KV-cache, autoregressive models can run much faster than diffusion and MAR. Moreover, reAR-B-AliTok achieves lower FID with faster sampling speed even against parallel-decoding approaches such as Maskbit, TiTok, VAR, and RandAR.

## 4.3 ABLATION STUDIES

We conduct ablation studies on the key components of reAR, focusing on the weighting and layer selection for encoding/decoding regularization, as well as the strategy for noise augmentation.

**Regularization Layer.** We analyzed the optimal layers for embedding regularization using reAR-S trained for 80 epochs without classifier-free guidance (Table 4). We ablated both the presence and placement of regularization and compared with the naive tied embedding strategy (Press & Wolf, 2016; Weber et al., 2024). For decoding regularization, early layers (e.g., layer 10) offer little benefit, while layer 15 performs best; applying it deeper slightly degrades performance. For encoding regulariza-

Table 3: **Ablation studies of noisy context regularization with annealing.**

| Noise Augmentation settings | FID ↓ |
|---|---|
| $\epsilon = 0.0$ | 2.12 |
| $\epsilon = 0.5$ | 3.15 |
| $\epsilon = 0.25$ | 2.08 |
| $\epsilon \sim U(0, 0.5)$ | 2.05 |
| $\epsilon \sim U(0, f(t)), \quad f(t) = 1 - t$ | 2.02 |
| $\epsilon \sim U(0, f(t)), \quad f(t) = \min(0, 1 - \frac{4}{3}t)$ | **2.00** |
| wo/ embedding regularization | 2.18 |

tion, the first layer is optimal as it aligns best with the token embeddings, whereas deeper layers harm generation quality. Notably, applying regularization to the layers closest to the target embedding space in vanilla AR yields the best results—encoding at layer 0 and decoding at roughly $\frac{3}{4}$ depth. We hypothesize this placement minimizes interference with next-token prediction. Based on these findings, we use EN@0 + DE@15 for reAR-S and EN@0 + DE@18 for reAR-B/L. We provide a more detailed comparison of different choices of the decoding regularization layer in Appendix C.2.

**Regularization Weight.** As shown in Table 4, regularization weight has a negligible impact on the quality of generation, likely because the AdamW optimizer is insensitive to the scale of the loss (Loshchilov & Hutter, 2017; Zhuang et al., 2022). For simplicity, we use $\lambda = 1$.

**Noise Augmentation.** We further ablate the design of noise augmentation, exploring two strategies: (1) assigning different noise levels to each token sequence, and (2) annealing the maximum noise level during training. Results are summarized in Table 3, based on the default setting with codebook embedding regularization (EN@0 + DE@15 for reAR-S). All models are trained for 400 epochs to evaluate the effect of different schedules. We find that a fixed noise level of $\epsilon = 0.25$ improves FID from 2.12 to 2.08, while a higher level ($\epsilon = 0.5$) leads to

Table 4: **Ablation studies of embedding regularization**. We use 'EN' as the encoding regularization and 'DN' as the decoding regularization. For example, 'DN@15' means applying decoding regularization at the 15th layer of the transformer block.

| Regularization settings | FID ↓ | IS ↑ |
|---|---|---|
| Vanilla AR | 21.32 | 57.3 |
| + tied codebook embedding | 21.08 | 57.2 |
| + DE@10 | 21.29 | 57.5 |
| + DE@15 | 20.03 | 61.0 |
| + DE@20 | 20.28 | 61.2 |
| + EN@0 + DE@20 | 19.83 | **61.7** |
| + EN@5 + DE@15 | 21.36 | 57.4 |
| + EN@0 + DE@15 (**Final choice**) | **19.72** | 61.3 |
| $\lambda := 0.5$ | 19.79 | 60.9 |
| $\lambda := 1.5$ | 19.74 | 61.5 |

training collapse (FID = 3.15). Randomizing the noise level within $[0, 0.5]$ further improves FID to 2.05. Incorporating an annealing schedule, where $f(t) = 1 - t$, yields a stronger result (2.02 FID). Finally, using a truncated linear schedule $f(t) = \max(0, 1 - \frac{4}{3}t)$ achieves the best performance of 2.00 FID. These results highlight the effectiveness of proper annealing noise augmentation.

**Joint Effect of Consistency Regularization.** As shown in Table 3, using only embedding regularization ($\epsilon$=0) yields an FID of 2.12, while using only noise augmentation yields 2.18. In contrast, combining the two further improves performance, reducing the FID of reAR-S to 2.00. This indicates that both noisy context regularization and codebook embedding regularization are important.

## 5 RELATED WORK

**Visual AR models** generate images by predicting pixels or patch tokens sequentially, each conditioned on previous context (Gregor et al., 2014; Van den Oord et al., 2016; Van Den Oord et al., 2016; Parmar et al., 2018; Chen et al., 2020). In this paper, we refer specifically to the visual AR model as the family using a unidirectional structure. Direct pixel-level modeling is expensive, so patch-based tokenizers (Van Den Oord et al., 2017; Esser et al., 2021) are used to compress local regions into discrete tokens. An AR model then predicts the token sequence (Esser et al., 2021; Sun et al., 2024; Luo et al., 2024). Prior work has focused on modular design, such as reducing quantization errors (Yu et al., 2023; Mentzer et al., 2023; Ma et al., 2025; Li et al., 2024) or exploring tokenization beyond standard 2D grids (Yu et al., 2024b; Miwa et al., 2025; Sargent et al., 2025; Xiong et al., 2025). Others have studied sequence dependencies, imposing causality during tokenizer training (Wu et al., 2025; Bachmann et al., 2025; Pan et al., 2025) or randomizing token order (Pang et al., 2025; Yu et al., 2024a). While these works focus on the flaw of a single component, we provide a novel perspective on the inconsistency between the AR model and the tokenizer.

**Other visual generation paradigm** has advanced from Variational Autoencoders (VAEs) (Kingma & Welling, 2013) and Generative Adversarial Networks (GANs) (Goodfellow et al., 2014) to modern approaches such as masked generative models (Chang et al., 2022; Yu et al., 2023; Weber et al., 2024) and diffusion-based models (Dhariwal & Nichol, 2021; Peebles & Xie, 2023; Ma et al., 2024; Yu et al., 2024c), apart from AR model. Recently, MAR (Li et al., 2024) was proposed to address quantization errors, and VAR (Tian et al., 2024) for next-scale prediction. However, they are not implemented in a decoder-only transformer, making them harder to incorporate with the standard AR used in large language models. We provide more discussion in Appendix D.

**Exposure bias** has been extensively studied in the language domain, with methods such as scheduled sampling (Bengio et al., 2015). In the visual domain, RQ-Transformer (Lee et al., 2022) applies scheduled sampling, and IQ-VAE (Zhan et al., 2022) uses Gumbel-softmax to mix ground-truth and predicted tokens, though both approaches compromise the parallel training efficiency of decoder-only Transformers. More recently, video generation works have addressed exposure bias in autoregressive diffusion models (Zhou et al., 2025; Huang et al., 2025), but these strategies are not applicable to discrete token prediction.

**Representation Alignment.** Representation alignment has been explored in visual generation (Yu et al., 2024c; Leng et al., 2025; Yao et al., 2025; Xiong et al., 2025). For example, REPA (Yu et al., 2024c) incorporates DINO-v2 features to accelerate diffusion training, and Disperse Loss (Wang & He, 2025) applies self-supervised objectives to improve diffusion representations. However, these methods are either designed for encoder-only Transformers and diffusion models or often rely on external visual encoders. In contrast, we aim to align the representations of the tokenizer and the AR model itself, requiring no external models and fitting naturally into the vanilla AR training pipeline.

## 6 CONCLUSION

In this paper, we identify the key bottleneck of visual autoregressive generation as the mismatch between the generator and the tokenizer, where the AR model struggles to produce token sequences that can be effectively decoded back into images. To address this, we propose reAR, a simple regularization method that substantially improves visual AR performance while remaining agnostic to tokenizer design. We hope this work will encourage future research on unifying generators and tokenizers within visual AR models, and more broadly, on developing unified multi-modal models.

## ACKNOWLEDGMENT

We would like to acknowledge that computational work involved in this research is partially supported by NUS IT's Research Computing group using grant number NUSREC-HPC-00001.

## ETHICS STATEMENT

Our work introduces a regularization strategy to improve visual autoregressive models and contributes toward the broader goal of unifying vision and language generation. While these advances can benefit research on unified multimodal models, we acknowledge the potential risks associated with generative technologies. In particular, improvements in fidelity and scalability may also lower the barrier for misuse, such as the creation of misleading or harmful synthetic media.

## REPRODUCIBILITY STATEMENT

We include all experiment details sufficient for reproducibility in Section. 4, Appendix A, Appendix B, and Appendix C. We provide the anonymous code here and will release the code once the paper is accepted.

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

# A  ADDITIONAL EXPERIMENTAL DETAILS

**Dataset and Evaluation Protocol.** For ImageNet evaluation, we follow the ADM protocol (Dhariwal & Nichol, 2021). Specifically, we compute both FID and IS using the ImageNet-1K validation split (50,000 images), and we generate 50,000 synthetic images with our model. We then compute FID between the generated set and the real validation set. During sampling, for classifier-free guidance, we adopt a power-cosine schedule as used in prior work (Zheng et al., 2023). For our reAR-S/B/L models, we set the guidance scale to 22, 14.5, and 10, respectively, and corresponding power scales to 2.75, 2.25, and 1.75. Across all models, we keep the temperature at 1.0 and do not use top-p or top-k sampling, so that improvements reflect model quality rather than sampling tricks. All the images generated and evaluated are fixed at the resolution of $256 \times 256$.

**Comparing methods.** We divide the visual generation into seven classes in Table 1: Diffusion model, MAR (continuous masked generation), Mask. (discrete masked generation), VAR (next scale prediction with encoder-only transformer), Rand. Causal AR (introduce randomized order of token sequence), Tok. Causal AR (use an advanced tokenizer that is not rasterization order), Raster. Causal AR (the most standard visual AR based on patch tokens and rasterization order).

**Model Configuration.** We use the same VQGAN tokenizer from MaskGIT (Chang et al., 2022), a pure CNN that produces feature maps which are patchified into $16 \times 16$ patches and quantized via a codebook of size 1024. For the autoregressive backbone, we follow the visual transformer (ViT)-based architecture of RAR (Yu et al., 2024a) and DiT (Peebles & Xie, 2023), further inserting class conditioning via AdaLN layers as in DiT. To ensure fair comparison with RAR, we use learnable positional embeddings throughout. We apply dropout with probability 0.1 both in the feed-forward network and in attention layers. Additionally, the MLP ratio is kept as 4.0 in the feed-forward network, and the number of attention heads is fixed to 16 for all different settings. We also include QK-Norm in attention to enhance stability.

**Training details.** As we mentioned in Section 4.1, all models are trained for 400 epochs with a batch size of 2048 on a single node of 8 A800 GPUs. For reAR-S and reAR-B, we use gradient accumulation as 1, and for reAR-L, we use gradient accumulation over 2 steps with a batch size of 1024 to achieve the same effective batch size. Following prior work (Yu et al., 2024a), we linearly warm up the learning rate to $4 \times 10^{-4}$ over the first 100 epochs and apply a

Table 5: **Comparison of computation cost.**

| Method | Time / Epoch (min) | FID |
|--------|--------------------|-----|
| AR-B   | 8.11               | 3.12 |
| reAR-B | 8.14               | 1.91 |
| AR-L   | 15.99              | 3.02 |
| reAR-L | 16.05              | 1.86 |

cosine decay schedule to decrease the learning rate to $1 \times 10^{-5}$ for the remaining 300 epochs. We use AdamW as the optimizer with $\beta_1 = 0.9$, $\beta_2 = 0.96$, and weight decay of 0.03. The gradient clipping is applied with a maximum gradient norm of 1.0. We use mixed precision with bfloat16 to accelerate training.

**Implementation details of reAR.** Regarding noisy context regularization, the noise ratio is sampled from a range that is determined by the training procedure. Specifically, the noise ratio is sampled from $(0, f(t))$, where $f(t) = \min(0, 1 - \frac{4}{3}t)$ and $t$ refers to the normalized training progress. For example,

Table 6: **Evaluation of reAR on VAR**

| Method | FID | IS |
|--------|-----|-----|
| VAR-d16 | 3.55 | 274.4 |
| VAR-d16 (retrained w/ reAR) | 3.39 | 276.6 |

the noise ratio is sampled from $(0, \frac{1}{2})$ at the 150 epoch where $t = \frac{3}{8}$ over total 400 epochs. Regarding codebook embedding regularization, the 2-layer MLP with hidden size as $2048$ is equipped with GeLU and maps the generator feature into the dimension of the corresponding codebook embedding. The parameter overhead of the MLP is 2.1M/2.1M/4.2M for reAR-S/B/L. Table 5 shows the training time cost on 8 A800 GPUs. This light-weight design only brings minimal training overhead while achieving superior performance with the same inference cost.

**Experiments on VAR (Tian et al., 2024).** VAR differs from standard autoregressive models such as VQGAN, TiTok, and AliTok, as it predicts the next scale or resolution and outputs multiple discrete tokens simultaneously rather than using a decoder-only transformer to predict a single next token. These differences lead to training and inference behaviors that diverge from standard AR, and we provide more details in Appendix D. Nevertheless, because VAR still generates discrete tokens autoregressively, it may also benefit from reAR. To test this, we apply reAR to VAR-d16 using the

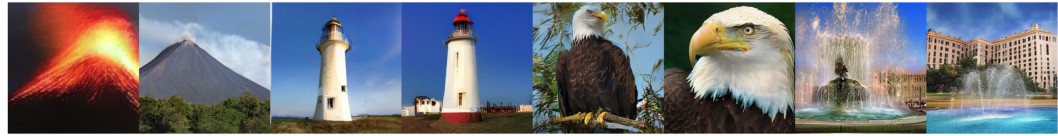

Figure 6: **Qualitative Results of VAR-d16 retrained with reAR.**

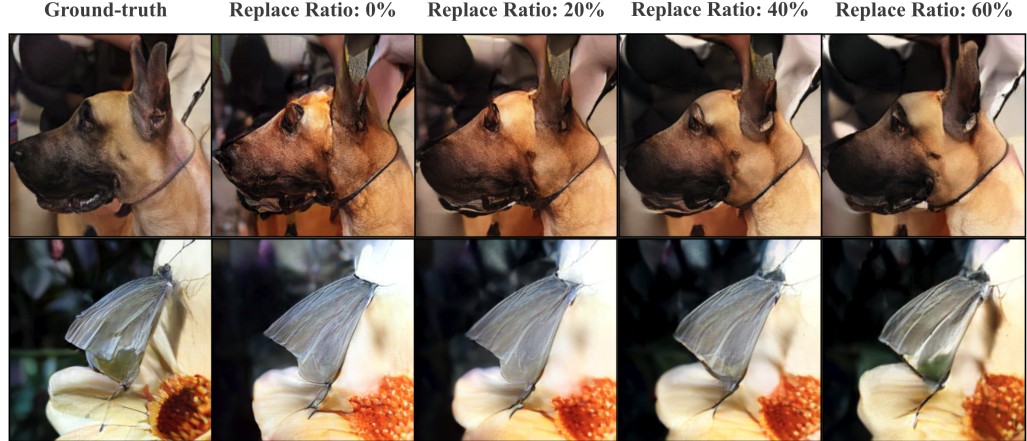

Figure 7: Visualization of analysis experiment on replacing tokens with more similar embedding.

same training settings as in our main experiments and same inference settings as in the original VAR paper. As shown in Table 6, reAR improves performance without tuning any training or inference hyperparameters, demonstrating its generalization ability.

## B    ANALYSIS DETAILS ON GENERATOR-TOKENIZER INCONSISTENCY

In this section, we present more details on the analysis experiments introduced in Section 3.1 on the generator-tokenizer inconsistency, including (i) evaluation metric (Section B.1), (ii) experiment settings (Section B.2), and (iii) Findings (Section B.3).

### B.1    EVALUATION METRIC FOR STUDYING INCONSISTENCY

We provide additional results on the quantitative evaluation of the inconsistency between token sequences $\mathbf{x}_{1:N}$ and the corresponding decoded images $\hat{\mathbf{I}}$. We adopt two groups of metrics: (i) for token sequence quality, we use the *correct token ratio* (CTR) and *perplexity*, and (ii) for image quality, we use PSNR and LPIPS. Here, the LPIPS and PSNR are different from those in the reconstruction task, since the decoded image is obtained from the generated token sequence under teacher forcing. While this is not a direct evaluation of the generation quality, it serves as an intermediate proxy similar to the correct token ratio and perplexity, but in pixel space.

**Evaluation on token sequence**. CTR measures the fraction of correctly predicted tokens under teacher forcing, while perplexity reflects the uncertainty of the predicted token distribution. Formally, given ground-truth sequence $\mathbf{x}_{1:N}$ and autoregressive model $p_\theta$, we define

$$\text{CTR} = \frac{1}{N} \sum_{i=1}^{N} \mathbf{1}\left[\arg\max_{v} p_\theta(v \mid \mathbf{x}_{1:i-1}) = x_i\right], \tag{8}$$

$$\text{Perplexity} = \exp\left(-\frac{1}{N} \sum_{i=1}^{N} \log p_\theta(x_i \mid \mathbf{x}_{1:i-1})\right). \tag{9}$$

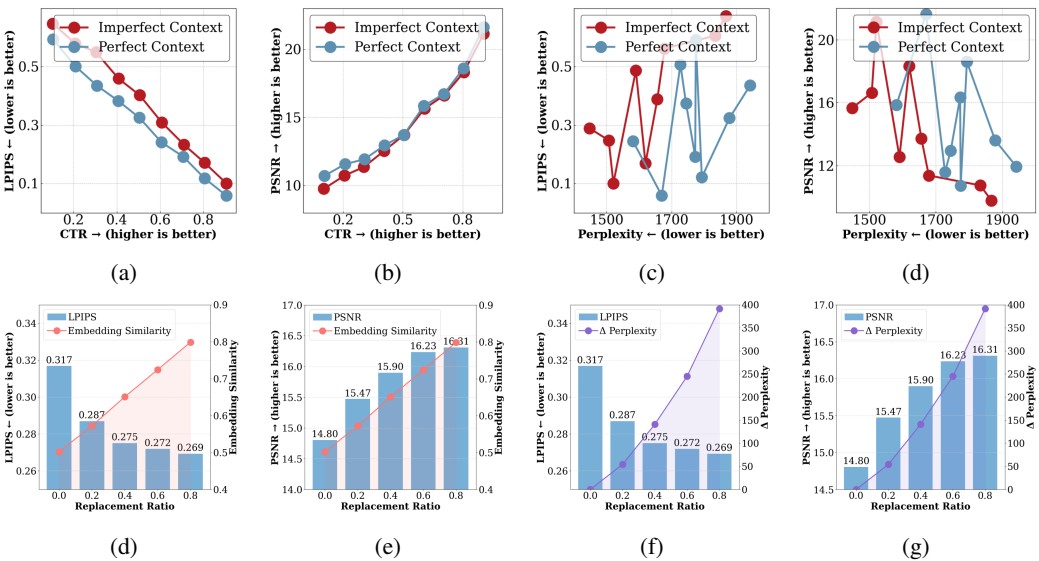

Figure 8: **Detailed analysis of generator–tokenizer inconsistency.** Results are evaluated with CTR (↑), perplexity (↓), PSNR (↑), and LPIPS (↓). (a–d): Exposure bias analysis—under the same CTR and lower perplexity, imperfect context yields higher LPIPS and lower PSNR. (e–g): Embedding unawareness analysis—Δ Perplexity denotes the increase from original to replaced sequences; even with similar CTR and lower perplexity, original predictions can give worse PSNR/LPIPS, showing that higher-quality token sequences can be decoded into worse images.

**Evaluation on decoded image**. To assess the quality of the decoded images $\hat{\mathbf{I}} = \mathcal{D}(\mathbf{z}^q)$, we report peak signal-to-noise ratio (PSNR) and learned perceptual image patch similarity (LPIPS). PSNR is a distortion-based metric that measures reconstruction fidelity relative to the ground-truth image $\mathbf{I}$:

$$\text{MSE} = \frac{1}{3HW} \sum_{c=1}^{3} \sum_{i=1}^{H} \sum_{j=1}^{W} \left( I_{cij} - \hat{I}_{cij} \right)^2, \tag{10}$$

$$\text{PSNR} = 10 \cdot \log_{10} \left( \frac{L^2}{\text{MSE}} \right), \tag{11}$$

where $L$ is the maximum possible pixel value (e.g., 255 for 8-bit images). A higher PSNR indicates better pixel-wise reconstruction fidelity.

LPIPS, on the other hand, evaluates perceptual similarity by comparing deep features extracted from a pretrained network $\phi$:

$$\text{LPIPS}(\mathbf{I}, \hat{\mathbf{I}}) = \sum_{l} \frac{1}{H_l W_l} \left\| w_l \odot \left( \phi_l(\mathbf{I}) - \phi_l(\hat{\mathbf{I}}) \right) \right\|_2^2, \tag{12}$$

where $\phi_l(\cdot)$ denotes the activation map from layer $l$, and $w_l$ are learned weights that calibrate the contribution of each layer. Lower LPIPS corresponds to higher perceptual similarity.

## B.2 ANALYSIS EXPERIMENT SETTINGS

To analyze the inconsistency between token sequence behavior and decoded image quality, we study the relationship between token-level metrics (CTR, Perplexity) and image-level metrics (LPIPS, PSNR). The key challenge is to design controlled interventions such that one aspect of quality (token sequence or image) can be varied while holding the other approximately fixed, thereby revealing causal effects. In all experiments, we treat *correct token ratio (CTR)* as the control variable, since it is the most straightforward to manipulate, while Perplexity, LPIPS, and PSNR serve as dependent variables. This setup allows us to investigate how changes in token correctness propagate to perceptual differences in reconstructed images.

**Experiments on amplified exposure bias.** As discussed in Section 3.1, we design two decoding protocols to vary the amount of exposure bias under the same CTR level:

- *Perfect Context (front-loaded):* Given a target CTR $r$, we fix the first $\lfloor rn \rfloor$ tokens to ground truth $x_{1:\lfloor rn \rfloor}$ and let the autoregressive model freely generate the remaining tokens. This minimizes exposure bias, since the context remains error-free until the switch point.

- *Imperfect Context (uniformly interleaved):* For the same CTR $r$, we randomly select $\lfloor rn \rfloor$ positions in the sequence and load ground-truth tokens only at those positions. At all other positions, tokens are sampled autoregressively. This introduces earlier corruption into the context and amplifies exposure bias.

Both settings guarantee the same number of ground-truth tokens, so any difference in downstream LPIPS/PSNR is attributable to the severity of exposure bias. This isolates the tokenizer's role in amplifying exposure bias during generation.

**Experiments on embedding unawareness.** While exposure bias focuses on *where* ground-truth tokens are inserted, embedding unawareness examines *what happens when incorrect tokens are replaced by semantically similar alternatives*. During training, the autoregressive model is optimized for exact token prediction, whereas the tokenizer decoder operates in a continuous embedding space. To study this gap, we introduce a replacement ratio $r' \in [0, 1]$:

1. First, generate predictions $\hat{x}_{1:n}$ with teacher forcing. Identify all positions $i$ where $\hat{x}_i \neq x_i$.

2. For each such incorrect prediction, replace $\hat{x}_i$ with probability $r'$ by another token $x'_i$ whose embedding $z^{q'}_i$ is the closest to the correct embedding $z^q_i$ under cosine similarity, i.e.,

$$z^{q'}_i = \arg\min_{z^q \in \mathcal{Z} \setminus \{z^q_i\}} d(z^q, z^q_i).$$

3. The CTR remains unchanged, since replacements are only among incorrect predictions, but the embedding similarity of the sequence is improved.

By varying $r'$, we control the degree of embedding similarity while holding CTR constant, and then measure its effect on LPIPS and PSNR of the reconstructed images. This design allows us to directly test whether embedding closeness—rather than token identity alone—affects perceptual quality.

**Summary.** For both experiments, we additionally evaluate the perplexity of the token sequence under the same CTR and study its correlation with LPIPS / PSNR as well. Together, these controlled settings—Perfect vs. Imperfect Context for exposure bias, and embedding replacement for unawareness—enable a systematic evaluation of how token-level inconsistencies translate into perceptual / pixel-level degradation in decoded images.

### B.3 FINDINGS AND OBSERVATION

**Results on exposure bias.** As shown in Figure 8(a–b), under the same CTR, sequences generated with *imperfect context* lead to higher LPIPS and lower PSNR, indicating worse decoded images, especially at low CTR. A similar trend is observed with perplexity. Although perplexity cannot be directly controlled, varying CTR naturally induces different perplexity levels. Thus, in Figure 8(c–d), we plot perplexity against PSNR/LPIPS under matched CTR. Even when the token sequence quality appears worse (higher perplexity), images decoded from tokens generated with *perfect context* still achieve better visual quality (lower LPIPS, higher PSNR) compared to those from *imperfect context*. This highlights that a token sequence favored by the autoregressive model does not necessarily yield a better decoded image.

**Results on embedding unawareness.** As shown in Figure 8(e–g), increasing the replacement ratio $r'$ improves embedding similarity while keeping CTR unchanged. This leads to consistent improvements in decoded image quality: LPIPS decreases and PSNR increases as more incorrect predictions are replaced with embedding-nearest tokens. Importantly, even though perplexity rises due to these replacements, the resulting images become visually closer to the ground truth. Figure 7 further illustrates this effect—images reconstructed from sequences with higher replacement ratios (20–60%) recover clearer object structures (e.g., sharper outlines of the dog's ears and the butterfly's wings)

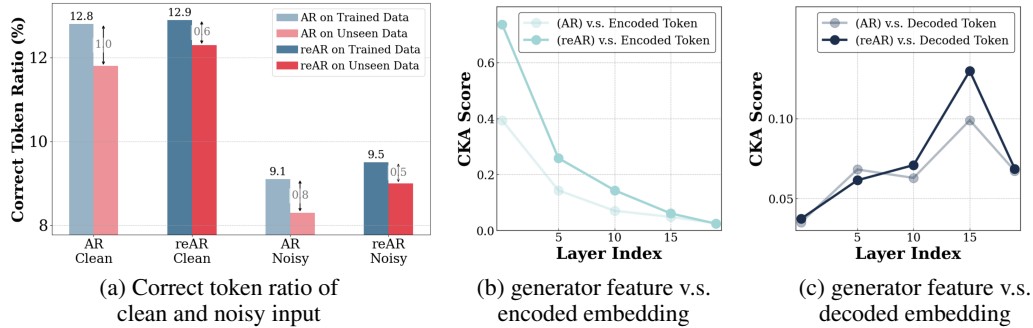

Figure 9: **Mitigating inconsistencies in visual autoregressive generation.** (a) reAR narrows the performance gap between trained and unseen data compared to vanilla AR and improves robustness under noisy inputs, indicating better generalization. (b, c) The CKA score demonstrates similarity between the feature and codebook embedding. reAR further aligns hidden features with the embedding of the current token in early layers and with the embedding of the next token in deeper layers.

compared to the 0% baseline. These results demonstrate that token correctness alone is insufficient to guarantee high-quality reconstructions; instead, embedding proximity plays a crucial role in aligning autoregressive predictions with tokenizer decoding.

## C    ANALYSIS ON THE EFFECT OF REAR

In this section, we present further analysis on the effect of reAR: (i) its effect on the token space (Section C.1) and (ii) its effect on the hidden features, which also includes the analysis on the choice of regularization layer as mentioned in Section 3.2 and Section 4.3.

### C.1    IMPACT ON SAMPLED TOKEN SEQUENCE

We found that reAR improves the next token prediction on: (i) generalization and (ii) robustness.

**Generalization.** We compare the correct token ratio (CTR) of vanilla AR and reAR on both trained data[1] and unseen validation data from ImageNet-1K as shown in Figure 9(a). On clean inputs, reAR achieves nearly identical performance to vanilla AR on trained data (12.9 vs. 12.8), but obtains higher CTR on unseen data (12.3 vs. 11.8), indicating improved generalization. These results suggest that incorporating codebook embeddings provides a stronger inductive bias for visual signals, enabling the AR model to learn more generalizable representations.

**Robustness.** To examine the robustness gained from reAR, we randomly replace a fraction of current tokens with noise at a controlled rate. Figure 9 (a) also compares the CTR for clean sequences and for sequences with 10% of tokens replaced uniformly. Compared to vanilla AR, reAR gains higher CTR compared to AR on the noisy trained data (9.5 v.s. 9.1). On the noisy and unseen data, the performance gap is even larger: reAR substantially outperforms vanilla AR (9.0 vs. 8.3). This result shows that reAR is more robust to the possible exposure bias.

### C.2    IMPACT ON HIDDEN FEATURES OF DIFFERENT REGULARIZATION LAYER

To better understand how reAR interacts with hidden representations, we evaluate the similarity between generator features and tokenizer embeddings using centered kernel alignment (CKA) (Kornblith et al., 2019). Specifically, given two sets of feature representations $\mathbf{X} \in \mathbb{R}^{n \times d_x}$ and $\mathbf{Y} \in \mathbb{R}^{n \times d_y}$, we first compute their Gram matrices $\mathbf{K} = \mathbf{X}\mathbf{X}^\top$ and $\mathbf{L} = \mathbf{Y}\mathbf{Y}^\top$, and then center them as $\mathbf{K}_c = \mathbf{H}\mathbf{K}\mathbf{H}$ and $\mathbf{L}_c = \mathbf{H}\mathbf{L}\mathbf{H}$, where $\mathbf{H} = \mathbf{I}_n - \frac{1}{n}\mathbf{1}_n\mathbf{1}_n^\top$ is the centering matrix. The

---

[1]To avoid class-wise bias, we sample 1000 images per class to match the validation setting.

CKA score is defined as

$$\text{CKA}(\mathbf{X}, \mathbf{Y}) = \frac{\langle \mathbf{K}_c, \mathbf{L}_c \rangle_F}{\|\mathbf{K}_c\|_F \|\mathbf{L}_c\|_F}, \tag{13}$$

where $\langle \cdot, \cdot \rangle_F$ denotes the Frobenius inner product and $\| \cdot \|_F$ is the Frobenius norm. Intuitively, CKA measures the alignment between the pairwise similarity structures of two representations and is invariant to isotropic scaling and orthogonal transformation. A higher CKA score indicates that the hidden features of the generator are more similar to the corresponding tokenizer embeddings.

**Analysis Target.** We aim to examine how hidden features within the decoder-only transformer correlate with two types of embeddings: the *encoded embedding* $\mathbf{z}_i^q$, representing the codebook vector of the current token, and the *decoded embedding* $\mathbf{z}_{i+1}^q$, corresponding to the codebook vector of the next token. By comparing generator features against both embeddings, we can assess how the autoregressive model's hidden representations evolve—capturing alignment with the tokenizer's codebook while simultaneously encoding the current token and preparing to decode the next one.

**Correlation between hidden features and embeddings.** To analyze how autoregressive representations evolve with depth, we compute CKA similarity between hidden features of a vanilla AR model and tokenizer embeddings across layers (Figure 9(b–c)). Four key trends emerge: (1) overall, CKA with the decoded embedding is lower than with the encoded embedding, since the current token is known while the next token remains uncertain; (2) similarity to the encoded embedding is highest at the input layer and decreases monotonically with depth; (3) similarity to the decoded embedding gradually increases and peaks around layer 15, roughly three-quarters of the full architecture; and (4) similarity to the decoded embedding

| Regularization settings | FID ↓ | IS ↑ |
|---|---|---|
| DE@13 | 20.47 | 59.4 |
| DE@14 | 20.17 | 60.8 |
| DE@15 | 20.03 | 61.0 |
| DE@16 | 20.11 | 60.5 |
| DE@17 | 20.25 | 61.1 |

Table 7: **Analysis on nearby regularization layer**. We use 'EN' as the encoding regularization and 'DN' as the decoding regularization. For example, 'DN@15' means applying decoding regularization at the 15th layer of the transformer block.

drops again in the final layers. Together, these patterns suggest a natural progression: early layers focus on encoding the current token and aggregating contextual information, while deeper layers shift toward modeling the next-token embedding. The decline in the final layers likely reflects the model's need to project features onto a decision boundary for prediction, where the codebook embedding itself may not form an optimal target. This also explains why directly tying AR outputs to codebook embeddings can lead to suboptimal performance.

**Choosing the regularization layer.** Motivated by these observations, we design reAR to apply regularization at layers where the CKA similarity is naturally high—early layers for encoded embeddings and later layers for decoded embeddings. Intuitively, this choice minimizes conflict with the primary next-token prediction objective, since these layers are already aligned with the tokenizer. Importantly, we avoid imposing regularization at the very last layer. Instead, we place regularization near the three-quarter depth of the model, where decoded embedding similarity is maximized. Empirically, we find that applying reAR to nearby layers yields similar performance as Table 7, highlighting the flexibility of our method with respect to the choice of regularization layer.

**Effect of reAR on feature alignment.** After introducing reAR, we observe consistent increases in CKA similarity between generator features and both encoded and decoded embeddings (Figure 9(b–c)) at the target layer. In early layers, reAR strengthens alignment with encoded embeddings, helping the generator encode current tokens similar to the tokenizer. In deeper layers, reAR improves similarity with decoded embeddings, ensuring that hidden features are better aligned with the next token. This result indicates that reAR directly improves the consistency between the hidden feature of the autoregressive model and the tokenizer.

## D    Additional discussion on the Related Work

In this section, we present a detailed discussion and comparison of related work. Diffusion models have achieved great success in many downstream visual tasks, including image editing (Nichol et al., 2022; Meng et al., 2022; He et al., 2024; Hertz et al., 2022) and personalized image generation (Gal et al., 2022; Ruiz et al., 2023; He & Yao, 2025; Tan et al., 2025). By contrast, visual autoregressive models are less frequently used in these domains, mainly because their generation quality often

lags behind that of diffusion models. A growing line of research aims to bridge this gap between visual autoregressive modeling and diffusion-based approaches. In the following, we mainly discuss that how these prior methods can be viewed through a unified lens: they address the inconsistency between the tokenizer (or tokenization scheme) and the autoregressive model. We also discuss how MAR and VAR differ from other autoregressive approaches, and highlight the distinction between our method and REPA, a regularization technique proposed for visual generation.

## D.1 TOKENIZATION WITH RANDOMIZED ORDER

**RandAR** (Pang et al., 2025) introduces a positional token in front of each patch token to let the token be aware of its position in terms of tokenization. Specifically, given a 256 token sequence, it inserts additional 256 tokens, and the generator is required to learn the distribution of the total 512 tokens under permutation. During training and inference, the token sequence is always shuffled. It enables parallel decoding during inference by inserting multiple positional tokens simultaneously. However, RandAR can double the context and significantly increase the computation budget.

**RAR** (Yu et al., 2024a) introduces a learnable embedding of target position over each token. During training, it randomly shuffles the token sequence at a given probability, and the token is aware of its own position with the additional positional embedding. It slowly decreases the probability of shuffling and returns to standard rasterization order during training. During inference, it keeps the standard operation for the autoregressive generation.

**Summary**. Both RandAR and RAR use permutation during training so that the context of each token is not limited to the tokens that are on the left or the top of it, thereby introducing bidirectional context even using a decoder-only transformer. This mitigates the inconsistency between the tokenizer that also models bidirectional context, such as MaskGiT-VQGAN (Chang et al., 2022) or TiTok (Yu et al., 2024b). However, in terms of the advanced tokenizer already introduced, unidirectional dependency, such as AliTok (Wu et al., 2025) and FlexTok (Bachmann et al., 2025), may further amplify the inconsistency as Table 2 in the main text shows.

## D.2 TOKENIZATION WITH 1D SEQUENCE OR UNIDIRECTIONAL DEPENDENCY

**TiTok** (Yu et al., 2024b) transforms an image into 1D discrete token sequence with query token using ViT. It firstly decouples the number of tokens from the number of patches and can further compress the number of tokens. However, the reconstruction quality of TiTok remains suboptimal compared to the patchify tokenizer. Additionally, although the represented token sequence is 1-dimensional, it's still in a bidirectional context instead of modeling unidirectional dependency. Therefore, the autoregressive model trained on it remains suboptimal as Table 2 in the main text shows.

**GigaTok** (Xiong et al., 2025) transforms the image into 1D discrete token sequence as well similar to TiTok. Additionally, it introduces the feature from DINO-v2, similar to REPA (Yu et al., 2024c) to regularize the hidden feature of the tokenizer decoder. This enables the tokenizer to scale up and stabilize training. However, it suffers from the same problem as TiTok, which still models bidirectional dependency.

**FlexTok** (Bachmann et al., 2025) firstly learns a continuous VAE with high fidelity. It then further resamples 1D discrete tokens from the 2D continuous token obtained from the VAE. Different from TiTok and GigaTok, it additionally employs a causal mask on the 1D sequence to model the unidirectional dependency, which is more consistent with an autoregressive model.

**AliTok** (Wu et al., 2025) introduces an Aligned Tokenizer that uses 1D sequences instead of the typical 2D patch grid, but with novel training to better align the tokenizer with autoregressive generation. Unlike standard patchified tokenizers, AliTok uses a causal decoder during tokenizer training to enforce unidirectional dependency among encoded tokens, so that tokens depend only on preceding ones. After that, it freezes the encoder and then uses a bidirectional decoder to refine the reconstruction quality. This unidirectional alignment improves compatibility with autoregressive models and leads to state-of-the-art generation metrics — our method still further enhances performance.

**Summary.** These works (e.g. TiTok (Yu et al., 2024b) and AliTok (Wu et al., 2025)) impose a 1D token sequence or enforce unidirectional dependency in the tokenization stage so that the tokenizer is more aligned with autoregressive models, which shows the importance of consistency between

the tokenizer and autoregressive model. In our experiments, we further demonstrate that using generator-tokenizer consistency regularization can further improve upon their performance.

### D.3 REMARKS ON MAR AND VAR

**MAR** (Li et al., 2024) is a model paradigm that combines masked prediction and autoregressive generation. Rather than generating tokens strictly in a raster (1D) order, MAR predicts multiple masked tokens in parallel across iterations, while still enforcing an ordering among iterations. Importantly, MAR uses continuous tokens instead of discrete ones and employs a diffusion-based head to model the continuous distribution of token predictions.

**VAR** (Tian et al., 2024) proposes a coarse-to-fine next-scale prediction strategy in image generation: rather than predicting each patch or token in a raster order, VAR generates images scale by scale, first at low resolution and then successively higher resolutions, where each finer scale is conditioned on all previously generated coarser scales. Given tokens of previous scale, the model will provide multiple mask tokens corresponding to the next scale, and decode them in parallel.

**Summary.** Although MAR and VAR can be regarded as autoregressive since generation proceeds in an autoregressive manner, they implement it with an encoder-only transformer or block causal transformer. In MAR, the model receives masked tokens as input and learns to reconstruct the masked positions, rather than predicting the next token in a decoder-only setup. In VAR, tokens from the previous resolution provide the context for predicting multiple tokens at the next resolution in parallel. Both model are different from standard AR paradigm of next token prediction.

### D.4 REGULARIZATION ON GENERATION TECHNIQUE

**REPA** (Yu et al., 2024c) is a regularization technique for diffusion-transformer models that aligns noisy intermediate states in the denoising process with clean image features from a pretrained visual encoder. Rather than forcing the model to learn image representations from scratch under noisy conditions, REPA adds a loss that encourages the hidden states of the diffusion model to match the semantic structure of an external teacher (e.g., DINO, DINO-v2).

**Comparison.** Unlike REPA, which focuses on accelerating the training of diffusion models, reAR is designed to address the inconsistency between autoregressive models and their tokenizers. Moreover, while REPA relies on external feature extractors such as DINO-v2 (Oquab et al., 2023), reAR directly leverages features from the tokenizer, which is already an integral component of the visual generation pipeline. In addition, REPA is tailored to bidirectional transformers and is restricted to 2D tokenizers, whereas reAR is compatible with decoder-only transformers. For these reasons, we do not apply REPA to visual autoregressive models, as it is less generalizable to visual AR training.

## E DISCUSSION ON THE LIMITATION

Our method has several limitations that suggest promising directions for future work. First, the choice of the decoding regularization layer is determined empirically. This issue is not unique to our approach, as prior works that regularize intermediate representations, such as REPA (Yu et al., 2024c) and Dispersive Loss (Wang & He, 2025), also depend on empirically selected layers in the absence of a clear theoretical principle. Developing an adaptive or theoretically grounded strategy for layer selection remains an open challenge and is more closely aligned with ongoing research in automated architecture and hyperparameter search.

Second, our experiments focus primarily on ImageNet, following common practice in foundational visual generative modeling (Esser et al., 2021; Yu et al., 2024b; Wu et al., 2025). While this setup enables controlled comparisons, we did not evaluate reAR on downstream text-guided generation tasks. A comprehensive evaluation on standard text-to-image benchmarks would offer a clearer assessment of practical utility, but is computationally demanding. We leave an expanded downstream study for future work.

Finally, although our empirical results demonstrate the effectiveness and generalization capability of reAR and we provide direct CKA analysis on the hidden feature of transformer layers before and after regularization, we do not provide a deeper theoretical analysis of the geometric factors

underlying generator–tokenizer alignment. Understanding properties such as manifold structure or distributional behavior could yield a more principled perspective, but developing such a theoretical framework is non-trivial. We view this as an important direction for future research.

## F    QUALITATIVE RESULTS

We present comprehensive generated results of reAR-B-AliTok (Figure 10 to 18) and reAR-L-VQGAN (Figure 19 to 24). All results are generated with a constant guidance scale of 4.0.

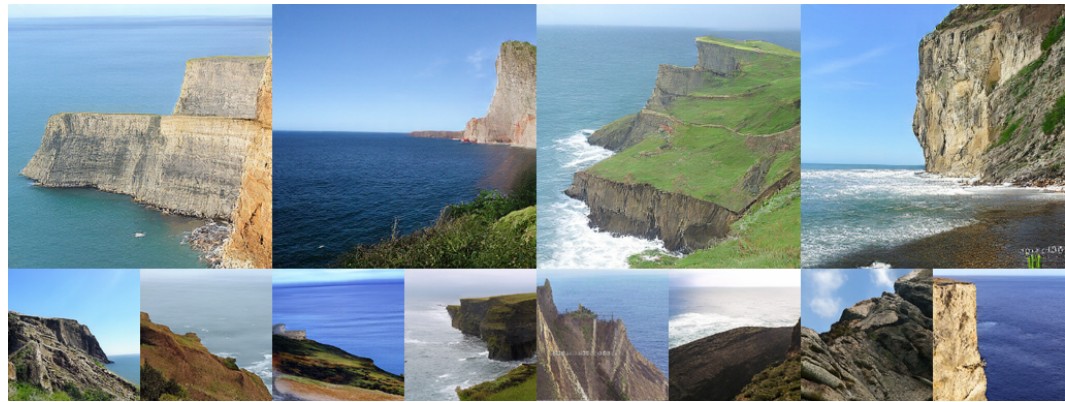

Figure 10: **Generated Results of reAR-B-AliTok of class 'Cliff'**

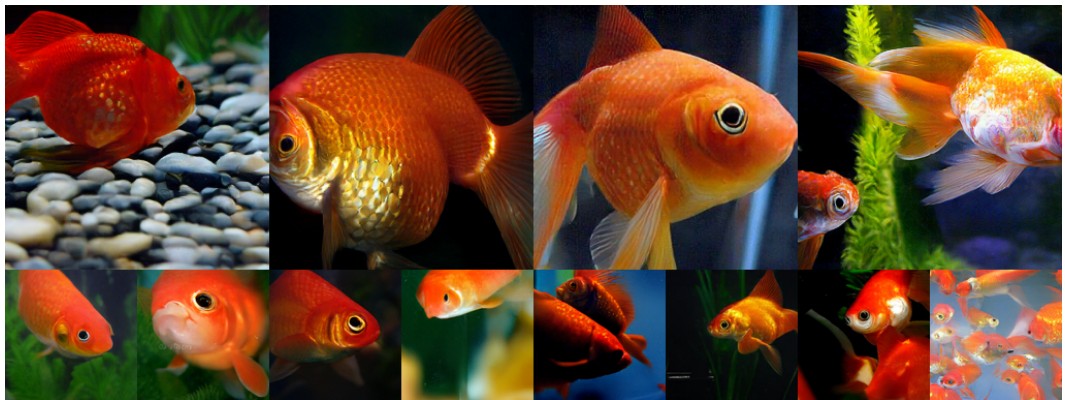

Figure 11: **Generated Results of reAR-B-AliTok of class 'Goldfish'**

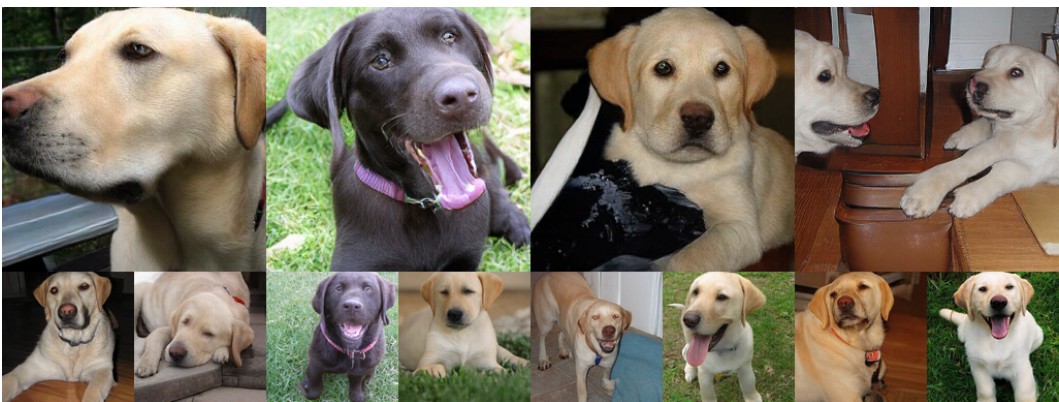

Figure 12: **Generated Results of reAR-B-AliTok of class 'Labrador retriever'**

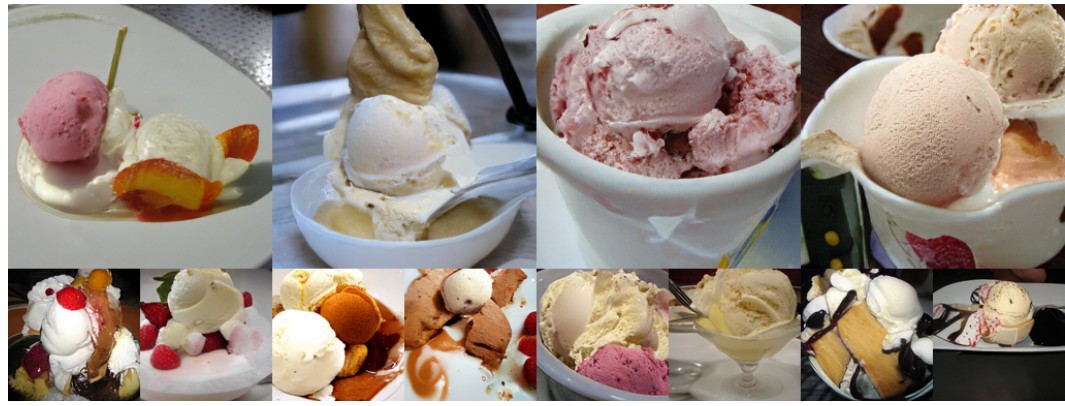

Figure 13: **Generated Results of reAR-B-AliTok of class 'Ice cream'**

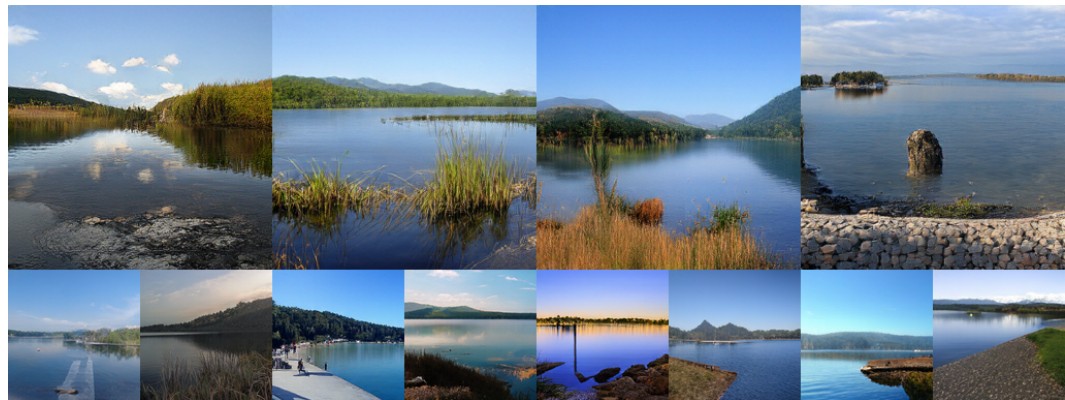

Figure 14: **Generated Results of reAR-B-AliTok of class 'Lakeshore'**

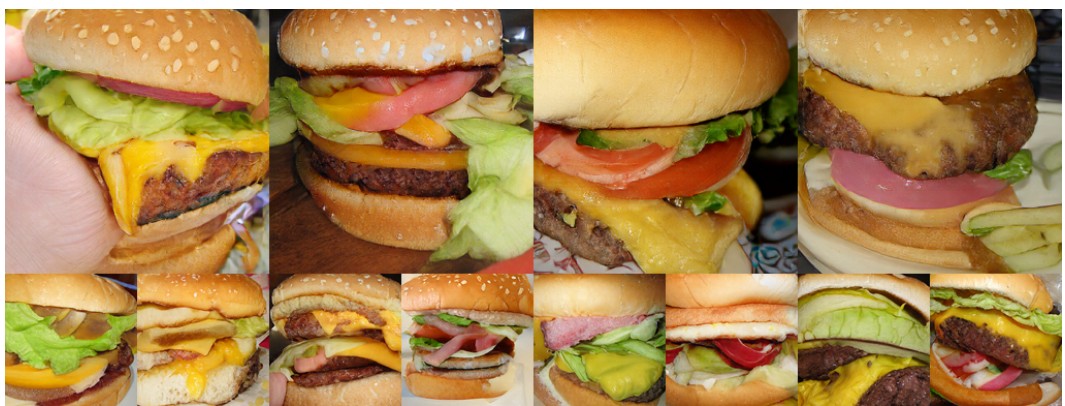

Figure 15: **Generated Results of reAR-B-AliTok of class 'Cheeseburger'**

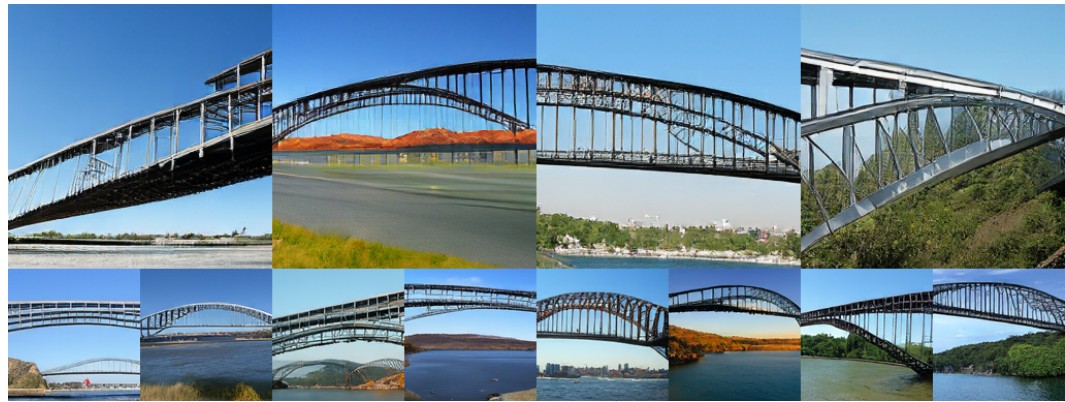

Figure 16: **Generated Results of reAR-B-AliTok of class 'Bridge'**

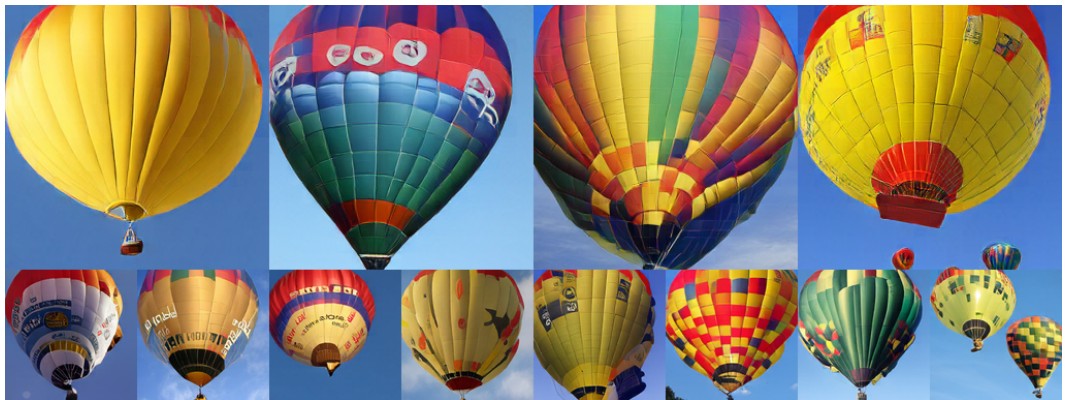

Figure 17: **Generated Results of reAR-B-AliTok of class 'Balloon'**

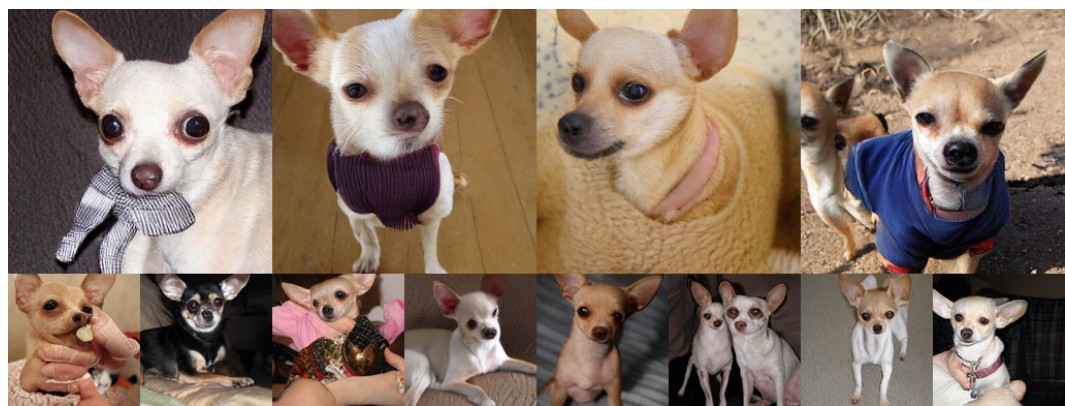

Figure 18: **Generated Results of reAR-B-AliTok of class 'Chihuahua'**

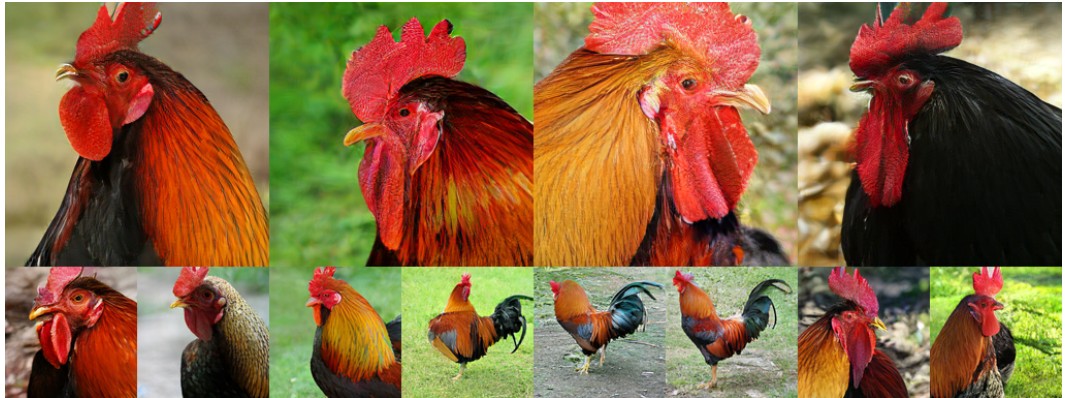

Figure 19: **Generated Results of reAR-L-VQGAN of class 'Cock'**

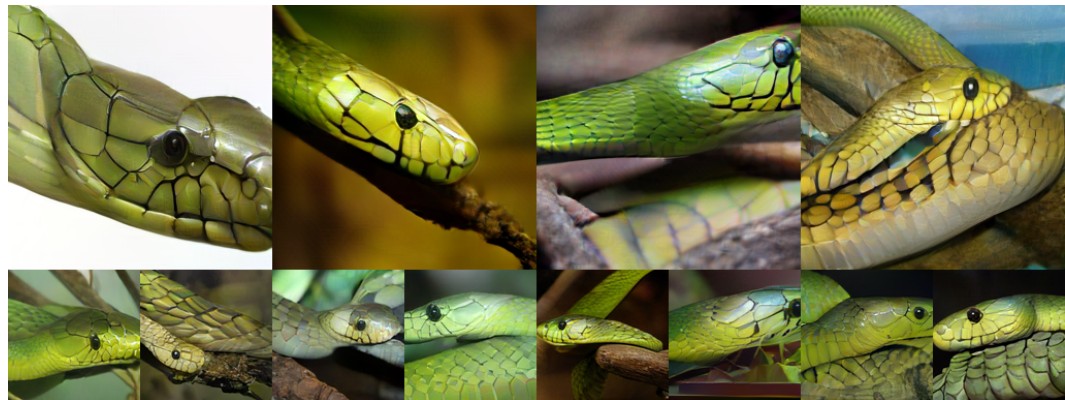

Figure 20: **Generated Results of reAR-L-VQGAN of class 'Green mamba'**

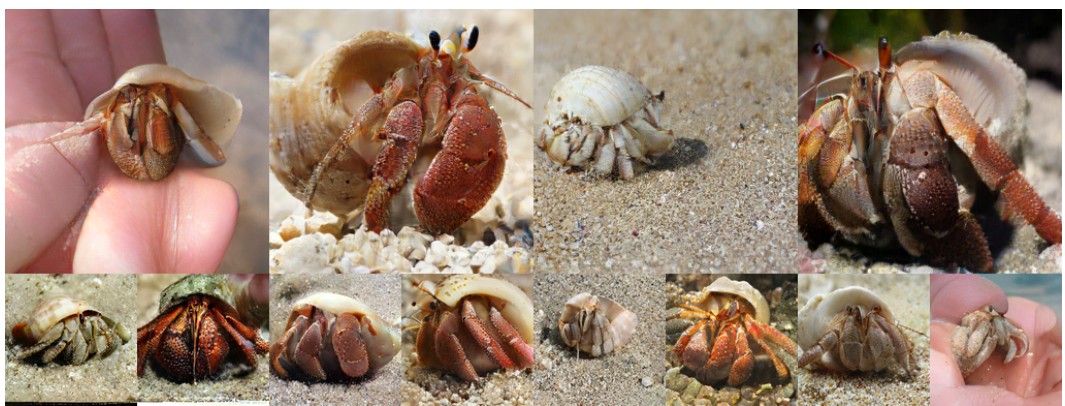

Figure 21: **Generated Results of reAR-L-VQGAN of class 'Hermit crab'**

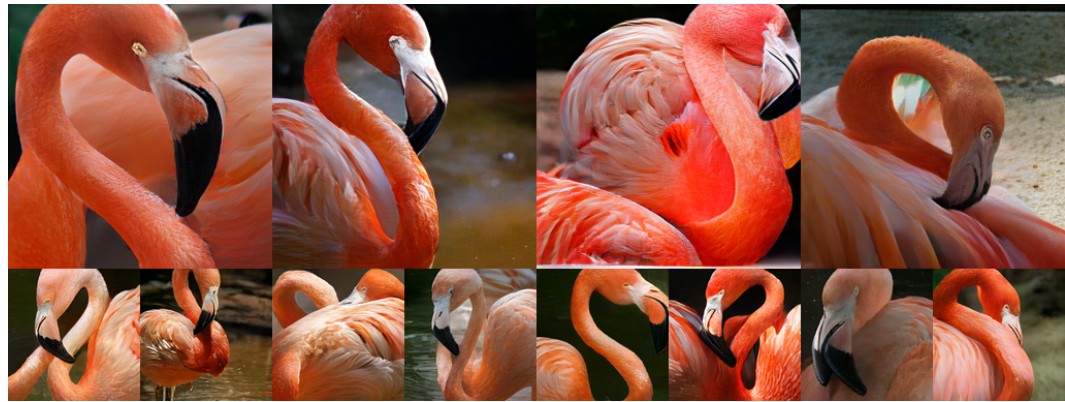

Figure 22: **Generated Results of reAR-L-VQGAN of class 'Flamingo'**

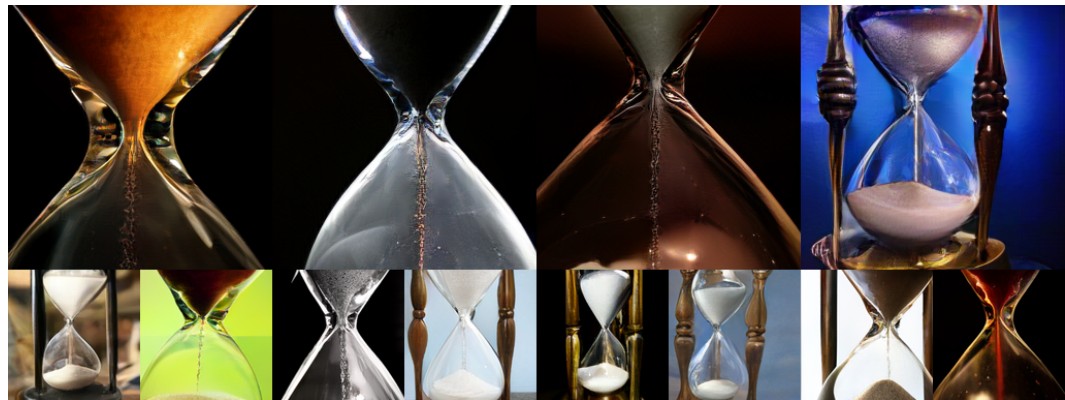

Figure 23: **Generated Results of reAR-L-VQGAN of class 'Hourglass'**

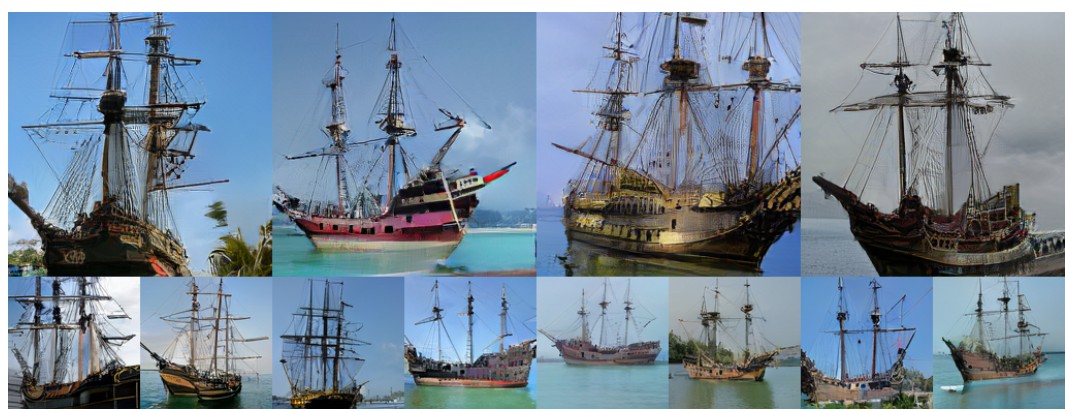

Figure 24: **Generated Results of reAR-L-VQGAN of class 'Pirate'**

