# OpenReview forum: "reAR: Rethinking Visual Autoregressive Models via Token-wise Consistency Regularization"
_ICLR.cc/2026/Conference — ICLR 2026 Poster_

### Official Review · Reviewer_RcEf · 2025-10-25

**Soundness:** 2
**Presentation:** 3
**Contribution:** 2
**Rating:** 4
**Confidence:** 4

**Summary:**

This paper proposes reAR, a new regularization framework designed to improve the generation quality of visual autoregressive (AR) models. The authors argue that existing visual AR methods suffer from a generator–tokenizer inconsistency, which manifests as exposure bias amplification and embedding unawareness during training and inference. reAR introduces two lightweight regularization techniques: Noisy Context Regularization, which injects random noise into the input token sequence to simulate imperfect contexts during training, mitigating exposure bias. Codebook Embedding Regularization, which aligns the generator’s hidden states with the tokenizer’s embedding space through cosine distance loss.

**Strengths:**

1. The paper identifies a genuine and underexplored problem: the misalignment between generator and tokenizer in visual AR models. This perspective is well-motivated and conceptually interesting.
2. The proposed regularizations are straightforward, computationally light, and compatible with existing AR models and different tokenizers.
3. The approach yields improvements across multiple tokenizers (e.g., TiTok, AliTok) and model sizes, and narrows the gap with diffusion-based methods.

**Weaknesses:**

1. While the paper frames generator–tokenizer inconsistency as a new perspective, the actual solutions (noise injection and embedding regularization) looks similar to existing techniques in language modeling.
2. For a method positioned as “plug-and-play” and intended to generalize across tokenizers and datasets, the paper applies the embedding alignment regularization to specific layers but does not provide analysis behind this choice. The decision appears empirical, based on trying a few configurations and selecting the one yielding the best FID.
3. While reAR demonstrates improvements across different tokenizers (e.g., TiTok, AliTok), the paper does not provide any analysis of how the proposed embedding regularization interacts with the underlying codebook geometry.

**Questions:**

1. How sensitive is reAR to the noise schedule or regularization strength λ across different datasets and architectures?
2. Since the proposed method involves embedding alignment, how does it interact with tokenizers of different codebook geometries?

---

> ### Author Response · Authors · 2025-11-21
>
> > While the paper frames generator–tokenizer inconsistency as a new perspective...
>
> Thank you for the thoughtful comment. Our core contribution is uncovering the fundamental generator–tokenizer inconsistency that arises uniquely in discrete visual autoregressive models. **Differnt from language modeling, the challenge is unique to visual AR models, because visual AR models rely on a separately trained discrete tokenizer, which creates a structural mismatch in their latent spaces.** Our method directly addresses this mismatch. The simplicity of the solution reflects the clarity and insight of the underlying problem rather than similarity to prior language-modeling techniques.
>
> We will appreciate explicit references on existing literature that propose similar techniques to help us enrich our discussion in our revision.
>
> > For a method positioned as “plug-and-play” ...
>
> We thank the reviewer for the suggestions on providing an analysis behind the choice.
>
> As we explained in Sec.. 3.2 (Codebook Embedding Regularization), our intuition is that the raw tokenizer embedding is not necessarily the optimal latent representation for next-token prediction. Applying regularization at a relatively deep layer allows the model to learn a tokenizer-informed representation and shift it to one that is more suitable for next-token prediction in the last several layers. Therefore, we choose a layer at 3/4 for each setting, which also demonstrates its effectiveness empirically. We also elaborate this in Appendix C.2 'Choosing the regularization layer'.
>
> Additionally, we would like to emphasize that reAR is relatively insensitive to the choice of the regularization layer. As shown in Table 3, the FID difference between EN@0 + DE@15 and EN@0 + DE@20 is small (19.72 v.s. 19.83). We further present Table 5 in our paper as follows:
>
> | Regularization Layer | FID |
> |---|---|
> | Baseline | 21.32 |
> | DE@13 | 20.47 |
> | DE@14 | 20.17 |
> | DE@15 | 20.03 |
> | DE@16 | 20.11 |
> | DE@17 | 20.25 |
>
> These results suggest that although the choice of regularization layer is a hyperparameter, the method is effective so long as the decoding-side regularization is applied at a sufficiently deep layer. Therefore, we recommend placing the regularization at approximately 3/4 of the total depth, a choice that works robustly across different architectures, including VQGAN, TiTok, and AliTok. To further verify robustness, we add results on VAR [1] with the same choice of regularized layer (i.e., layer 12 of VAR-d16) and regularization weight:
>
> | Method | FID | IS |
> |---------|----------|----------|
> | VAR-d16 | 3.55  | 274.4  |
> | VAR-d16 retrained with reAR | 3.39 | 276.6  |
>
> Prior works that regularize intermediate features (REPA [2], Dispersive Loss [3]) also rely on empirically chosen layers. Similar to learning-rate schedules or architectural design choices in deep networks, developing an adaptive or theoretically optimal mechanism for layer selection is highly challenging and is more aligned with ongoing research in AutoML.
>
> As such, we provide empirical analysis on the feature similarity of the codebook embedding using the CKA score (see Appendix C.2). The CKA score trend is aligned with our intuition: in the original model, the hidden feature became closer to the embedding of decoded token and decrease after a certain depth, which indicates that imposing regularization at approximately 3/4 depth is a reasonable choice. We hope that these could address your concern regarding the choice of the regularization layer.

---

> > ### Author Response · Authors · 2025-11-21
> > **Continue**
> >
> > > How sensitive is reAR to the noise schedule or regularization strength λ across different datasets and architectures?
> >
> > reAR is not sensitive to the choice of noise schedule and regularization strength across different architectures. We provide a comprehensive ablation study on the noise schedule in Table 4 and the regularization strength and Table 3. Our experiments on TiTok, AliTok, and VAR adopt the same configuration, including $\lambda = 1$, regularized encoding  layer at the first, regularized decoding layer at 3/4 depth, and annealing noise schedule. We also provide ablations on AliTok trained for 400 epochs using the default settings and without classifier-free guidance:
> >
> > | Method | FID |
> > |---|---|
> > | AliTok-reAR+ep=0.25 | 3.08  |
> > | AliTok-reAR+ep~U(0, 5) | 3.05 |
> > | AliTok-reAR+ep~U(0, f(t)), f(t) = min(0, 1 - 4/3t) | 3.01 |
> >
> > Additionally, we use $\lambda=1$ as the default setting across different architectures, which shows effectiveness across different architectures as well. As we discussed in Regularization Weight in Section 4.3 and presented in Table 3, the AdamW optimizer makes it less sensitive to the specific scale of $\lambda$ at the same order of magnitude.
> >
> > > While reAR demonstrates improvements across ... codebook geometry.
> >
> > > Since the proposed method involves embedding alignment...
> >
> > We thank the reviewer for raising this important point. We agree that understanding how embedding regularization interacts with codebook geometry is valuable. We have detailed analyses in Appendix C.2, the results of which are summarized in Section 3.2 (lines 308-314). Specifically, we use a CKA-based metric to measure geometric similarity between hidden states and codebook embeddings. As shown in Fig. 8(b,c), CKA scores increase markedly at the regularization layer, indicating that reAR indeed drives hidden features to become more aligned with the underlying codebook space.
> >
> > A deeper investigation of geometric factors—such as manifold curvature or distributional properties—would certainly be interesting, but is non-trivial.  We view it as an important direction for future work. We thank the reviewer again for the insightful suggestion.
> >
> > **Again, we thank the reviewer for all the valuable suggestions and updated the revision accordingly. We welcome further questions and comments for improving our submission.**
> >
> > [1] VAR, NeurIPS 2024.
> >
> > [2] REPA, ICLR 2025.
> >
> > [3] Dispersive Loss: Image Generation with Feature Regularization, arXiv 2025.

---

> > > ### Comment · Reviewer_RcEf · 2025-11-25
> > >
> > > Thanks for your detailed reply, I will raise my score to 6.

---

### Official Review · Reviewer_n9Nt · 2025-10-29

**Soundness:** 3
**Presentation:** 3
**Contribution:** 3
**Rating:** 6
**Confidence:** 3

**Summary:**

This paper analyzes the performance bottleneck in visual autoregressive (AR) models, attributing it to "generator-tokenizer inconsistency." To address this, the authors propose reAR, a plug-and-play training regularization strategy. Without altering the model architecture or inference pipeline, reAR introduces two auxiliary objectives: 1) Noisy Context Regularization, which mitigates exposure bias by training the model in corrupted contexts, and 2) Codebook Embedding Regularization, which forces the model's hidden states to predict the visual embeddings of both the current and next tokens, making the generator aware of the tokenizer's embedding space.

**Strengths:**

1. reAR delivers excellent results, significantly improving model performance without requiring architectural changes, and appears to be a general-purpose strategy for vision AR models.
2. As a training-only strategy, reAR is highly versatile. Experiments show it works well not only with VQGAN but also significantly boosts other tokenizers like TiTok and AliTok.
3. The paper clearly defines the "generator-tokenizer inconsistency" problem. It uses well-designed comparison experiments (e.g., "perfect context" vs. "imperfect context" and "error token embedding replacement") to convincingly validate that "exposure bias amplification" and "embedding-unawareness" are indeed critical bottlenecks.

**Weaknesses:**

1. reAR requires applying regularization at specific shallow (e.g., layer 0) and deep (e.g., layer 15) layers, chosen based on CKA analysis and ablations. This feels somewhat like "alchemy" or fine-tuning, lacking an adaptive or theoretically-driven mechanism to automatically determine the optimal layers for feature alignment.
2. The method introduces additional MLP projection layers and an extra loss term, which inevitably increases training complexity and memory consumption. The paper doesn't explicitly quantify the additional training time overhead introduced by the reAR strategy.

**Questions:**

1. reAR innovatively regularizes both the "current token" embedding (shallow layers) and the "next token" embedding (deep layers). Is there a potential conflict between these two objectives? (e.g., could the shallow layers lose information needed to predict the *next* token in their effort to align with the *current* token?).
2. Could you provide an ablation study showing that only regularizing the "next" token's embedding (which seems like the more direct objective) performs worse than regularizing both?

---

> ### Author Response · Authors · 2025-11-21
>
> > reAR requires applying regularization at specific shallow (e.g., layer 0) and deep (e.g., layer 15) layers, chosen based on CKA analysis and ablations. This feels somewhat like "alchemy" or fine-tuning, lacking an adaptive or theoretically-driven mechanism to automatically determine the optimal layers for feature alignment.
>
> We thank the reviewer for raising this point.  Our methodology builds on the observation that aligning the decoder’s hidden states with the tokenizer’s representation space is more important than the exact choice of the layer at which the alignment is applied; thus, the specific regularization layer serves primarily as an effective yet flexible design choice rather than a sensitive hyperparameter. We would like to emphasize that reAR is relatively insensitive to the choice of the regularization layer. As shown in Table 3, the FID difference between DE@15 and DE@20 is small (0.6%). We further present Table 5 in our paper as follows:
>
> |Regularization Layer|FID|
> |---|---|
> |Baseline|21.32|
> |DE@13|20.47|
> |DE@14|20.17|
> |DE@15|20.03|
> |DE@16|20.11|
> |DE@17|20.25|
>
> These results suggest that although the choice of the regularization layer is a hyperparameter, the method remains consistently effective, so long as the decoding-side regularization is applied at a sufficiently deep layer. In the paper, we recommend placing the regularization at approximately 3/4 of the total depth, a choice that works robustly across different tokenizers, including VQGAN, TiTok, and AliTok and model scale, including reAR-S, reAR-B, and reAR-L.
>
> Our intuition is that the raw tokenizer embedding is not necessarily the optimal latent representation for next-token prediction. Applying regularization at a relatively deep layer allows the model to learn a tokenizer-informed representation and shift it to one that is more suitable for next-token prediction in the last several layers. Therefore, we choose a layer at 3/4 of the overall depth by default. The empirical results on several architectures also reflect the effectiveness and flexibility of the choice of a specific layer.
>
> Finally, prior works that regularize intermediate features [1,2] also rely on empirically chosen layers. Similar to learning-rate schedules or architectural design choices in deep networks, developing an adaptive or theoretically optimal mechanism for layer selection is highly challenging and is more aligned with ongoing research in AutoML.
>
> > The method introduces additional MLP...
>
> Thanks for your valuable comment. We provide details using reAR-B trained on 8 A800 GPUs as an example:
>
> | Method | Training Time (mins) Per Epoch |FID|
> |---|---|---|
> |AR-B|8.11|3.12|
> |reAR-B|8.14|1.91|
> |AR-L|15.99|3.02|
> |reAR-L|16.05|1.86|
>
> Since we adopt a light-weighted MLP project layer, the overhead, including training time and memory cost (parameters + activation), is relatively small (0.3%). Quantitatively, training over 400 epochs only requires an additional 12 minutes to gain a large improvement over FID while maintaining the same inference cost. We have added the detailed training overhead in the revision following your suggestions.
>
> > reAR innovatively regularizes ...
>
> Thank you for this insightful question! Table 3's ablation study shows that removing the current-token embedding regularization at the first layer leads to a 0.45 degradation in FID, indicating that this component is indeed beneficial. We agree that if the current-token embedding were regularized at deeper layers (e.g., layer 5), there could be interference with the model’s ability to propagate information needed for next-token prediction.
>
> For this reason, we intentionally place the current-token regularization only at the first layer, before the autoregressive model begins aggregating context through attention. This design ensures that (1) the tokenizer-encoded prior is injected early, and (2) no downstream information required for predicting the next token is suppressed or overwritten. In practice, we find that this placement avoids the potential conflicts while providing consistent performance gains. We also added the clarifications to the revision.
>
> > Could you provide an ablation study ...?
>
> This study corresponds to DE@15 and DE@20 of Table 3, which we reorganize below. Only regularizing the "next" token reduces the performance compared to EN@0 + DE@15 and EN@0 + DE@20.
>
> | Regularization Layer | FID |
> |---|---|
> | Baseline | 21.32 |
> | DE@20 | 20.28 |
> | EN@0 + DE@20 | 19.83 (-0.45) |
> | DE@15 | 20.03 |
> | EN@0 + DE@15 | 19.72 (-0.31) |
>
> Hence, regularizing current token embedding is also important.
>
> **Again, we thank the reviewer for all the valuable suggestions and updated the revision accordingly. Please let us know if there are any additional questions we can help address.**
>
> [1] Representation Alignment for Generation: Training Diffusion Transformers Is Easier Than You Think, ICLR 2025.
>
> [2] Dispersive Loss: Image Generation with Feature Regularization, arXiv 2025.

---

> ### Comment · Reviewer_n9Nt · 2025-11-26
>
> The authors have fully addressed my concerns, and I have decided to improve my rating.

---

### Official Review · Reviewer_Xk7f · 2025-11-01

**Soundness:** 3
**Presentation:** 3
**Contribution:** 3
**Rating:** 6
**Confidence:** 4

**Summary:**

This work identifies a key bottleneck in visual autoregressive generation—the mismatch between the generator’s token sequences and how the tokenizer decodes them. The authors propose a plug‑and‑play regularization strategy that forces the autoregressive model to align its hidden representations with the tokenizer’s codebook embeddings and to remain robust under noisy contexts. This approach requires no changes to the tokenizer, inference pipeline or generation order, yet yields substantial gains. The method is validated across different tokenizers, showing that improving generator ‑ tokenizer consistency significantly boosts AR image generation.

**Strengths:**

1. The paper clearly identifies a fundamental issue in visual autoregressive models — the inconsistency between the generator and tokenizer — and presents a well-motivated solution.

2. The proposed generator–tokenizer consistency regularization is simple, plug-and-play, and does not require changes to the tokenizer, inference pipeline, or generation order, making it broadly applicable.

3. Extensive experiments demonstrate that reAR significantly enhances class-conditional image generation, yielding notable improvements in FID on ImageNet.

**Weaknesses:**

1. The paper does not report the impact of REAR on downstream multimodal understanding tasks or text-to-image generation benchmarks, leaving the broader utility of the method unclear.

2. While the method improves generation quality, the novelty is somewhat limited, as the approach mainly introduces a regularization term rather than a fundamentally new architecture.

3. The study only investigates reAR’s generalization across different VQ tokenizers, but does not examine its effectiveness when applied to different LLM backbones; additionally, the number of VQ tokenizers included in the experiments is limited.

**Questions:**

N/A

---

> ### Author Response · Authors · 2025-11-21
>
> We sincerely thank the reviewer for the careful and constructive review. We are happy that the reviewer thinks our paper "identifies a fundamental issue and proposes well-motivated solutions with notable empirical improvements". Below, we address your concerns separately.
>
> > The paper does not report the impact of REAR on downstream multimodal understanding tasks or text-to-image generation benchmarks, leaving the broader utility of the method unclear.
>
> We thank the reviewer for pointing this out. In this paper, we follow the common experimental settings adopted in prior studies on foundational visual generative models [1,2,3].  These methods evaluate only on image classification, on ImageNet. We greatly appreciate your suggestion to apply our methods to the downstream benchmarks. Due to the limited computation resources, we will investigate on multimodal udnerstanding and text-to-image generation in future works. We have added the related discussion in the revision.
>
> > While the method improves generation quality, the novelty is somewhat limited, as the approach mainly introduces a regularization term rather than a fundamentally new architecture.
>
> We thank the reviewer for the comment. We respectfully note that novelty is not limited only to architectural innovations.  Our aim is to address a fundamental and previously underexplored flaw in discrete autoregressive (AR) generation, i.e., the generator-tokenizer inconsistency. This inconsistency affects all discrete AR models, regardless of architectural design. This universality underscores the novelty and importance of our contribution.
>
> As for the approach, the **choice of a simple and architecture-agnostic solution is intentional**.  This allows it to be readily integrated into existing training pipelines and infrastructures in a plug-and-play manner. We believe that this practicality and broad applicability are advantages that will highlight the contribution and impact of our method.
>
> > The study only investigates reAR’s generalization across different VQ tokenizers, but does not examine its effectiveness when applied to different LLM backbones; additionally, the number of VQ tokenizers included in the experiments is limited.
>
> We thank the reviewer for the suggestion to evaluate the method on additional LLM backbones and VQ tokenizers. Our experiments feature three distinct tokenizers (VQGAN, TiTok, and AliTok), which represent, respectively, (i) a standard rasterization-order tokenizer, (ii) a 1D tokenizer, and (iii) a more sophisticated 2D-prefix design. These choices were made to cover a broad spectrum of commonly used discrete visual tokenizers. Furthermore,  the backbones used in these settings are also different.
>
> 1. In reAR + VQGAN, the backbone follows the VAR-style design, incorporating AdaLN but without RoPE.
> 2. In reAR + AliTok, the backbone resembles a standard LLM-style decoder, i.e., without AdaLN but with RoPE.
>
> The consistent improvements across these heterogeneous tokenizers and backbones provide empirical evidence of reAR’s broad applicability. In addition, following your suggestions, we further include results on VAR [1], which introduces yet another combination of tokenizer and backbone:
>
> | Method | FID | IS |
> |---------|----------|----------|
> | VAR-d16 | 3.55  | 274.4  |
> | VAR-d16 retrained with reAR | 3.39 | 276.6  |
>
> Hyperparameters were not tuned due to time constraints, yet the gains remain clear. These results further support the generalization ability of our method across diverse tokenizer–backbone configurations.
>
> **Again, we thank the reviewer for all the valuable suggestions and updated the revision accordingly. We welcome further questions and comments to improve our submission.**
>
> [1] Visual Autoregressive Modeling: Scalable Image Generation via Next-Scale Prediction, NeurIPS 2024.

---

> > ### Author Response · Authors · 2025-11-27
> >
> > Dear Reviewer Xk7f,
> >
> > We sincerely appreciate your suggestions on clarifying our contribution and expanding the experiments to additional base models, including VAR. As the discussion period ends in a few days, we would be glad to provide any further details that may help address the remaining concerns.
> >
> > Thank you again for your time and helpful suggestions!
> >
> > Best regards,
> >
> > The Authors

---

### Official Review · Reviewer_YZ8R · 2025-11-03

**Soundness:** 3
**Presentation:** 2
**Contribution:** 3
**Rating:** 6
**Confidence:** 4

**Summary:**

This paper tries to improve (traditional, raster-order) autoregressive image generation.

Problem statement: inconsistency between a generator and a tokenizer, i.e., the autoregressive model might generate a token sequence that is hard for the tokenizer to decode back to an image.

Logical development
* The generated token sequence can be unseen by the tokenizer due to exposure bias between training (teacher forcing) and inference (own predictions). It is more problematic in image models than language models because the possibility of the decoder not seeing the combination of generated visual tokens in the training phase.
* AR models produce token indices and do not care the embeddings produced by the tokenizer.


Solutions
* Expose the model to perturbed context during training.
    * (context = previous tokens)
    * It encourages the model not to rely on clean contexts and improves robustness to imperfect histories at inference.
* Align the generator's hidden states with the tokenizer's embedding space.
    * It helps the visual embeddings being reconstructed from the unseen tokens.
    * Method: SimSiam-like architecture and loss for regressing current embedding and next embedding.

Experiments
* Competitors: many.
* ImageNet
* FID 1.42 with 177M parameters (sota diffusion models = 675M)

**Strengths:**

Originality:
1. The idea of aligning the embeddings of the generator and the tokenizer is original.
2. Please see weakness 1.

Quality:
1. The experiments thoroughly compare many competitors in Table 1.
2. Please see weakness 2.
3. Ablation study is thorough.

Clarity:
1. The problem statement, logical development, solution, and empirical supports are clear as described in Summary.
2. Please see weakness 3.

Significance:
1. The inconsistency between the embeddings of generator and tokenizer and its solution are important because they are the fundamental components of AR models.

**Weaknesses:**

1. Perturbing something is a common practice for robustness. Why is the proposed perturbation non-obvious compared to the literature?
2. Experiments are conducted only on ImageNet.
3. The scope should be specified in more detail such as raster-order autoregressive modeling because the paper does not tackle other autoregressive models such as [Visual Autoregressive Modeling: Scalable Image Generation via Next-Scale Prediction].

minor

1. The term "context" should be defined. It is okay-ish to be implied from context (lol) but it can be clearer.

**Questions:**

1. Figure 1 should be explained in more detail. What is wrong in the image of orange cat? Why are the top and bottom images similar although the token indices are different?
2. Resolving weakness 1 will raise my rating regarding originality.
3. Resolving weakness 2 will raise my rating regarding soundness.
4. Resolving weakness 3 will raise my rating regarding clarity. If the proposed method is applicable to VAR, please help me find the statement in the paper.

---

> ### Author Response · Authors · 2025-11-21
>
> We thank the reviewer for the positive and comprehensive evaluation! We respond to each point and provide small edits to improve the paper.
>
> > Perturbing something is a ... to the literature?
>
> We agree that perturbation is a common strategy for robustness. However, our method is not for addressing robustness but generator-tokenizer inconsistency. Perturbation implemented in reAR is different from existing methods in both motivation, which we elaborate below:
>
> **Motivation:** To the best of our knowledge, existing visual AR frameworks [1,2,3] have largely overlooked the exposure bias introduced by the tokenizer, where unseen token sequences degrade the image decoding. As such, they do not apply any perturbation during training. Our perturbation is motivated by recovering the current and next embeddings in robust way to improve generator–tokenizer consistency.
>
> **Implementation:** While perturbation is common practice for robustness, it's commonly used for resolving exposure bias. Previous approaches that address exposure bias [4,5,6] rely on free-running decoding.  Such approaches require multiple forward passes per sequence and are typically used in post-training or small-scale setups. In contrast, our method applies uniform noise with an annealing schedule under teacher forcing, requiring only a single forward pass while achieving substantial quality gains. This simplicity makes it both scalable and practical for large-scale visual AR models.
>
> > Experiments are conducted only on ImageNet.
>
> We follow the experimental settings of prior works on pretraining methods of visual generative models [1,2,3]; they also evaluate exclusively on ImageNet. To test for generalization, we test over several tokenizers, including VQGAN, TiTok, AliTok, etc. We also note that each experiment on ImageNet is quite computationally intensive, requiring ~800 GPU-hours. We greatly appreciate your suggestion to expand to additional datasets, and we plan to explore the mechanism of reAR on standard text-to-image benchmarks in future work. We have added the related discussion in the revision.
>
> > The scope ... via Next-Scale Prediction].
>
> Thanks for this suggestion. Actually, our method is not limited to only raster-order autoregressive modeling. It is applicable to all discrete autoregressive generation using standard decoder-only Transformers (see Line 1 of the main paper and elaboration in Appendix D). For instance, both TiTok (1D) and AliTok (2D with prefix tokens) fall outside the conventional raster-scan paradigm, yet remain fully compatible with our approach.
>
> We focus on standard decoder-only Transformer AR models, since it is compatible with language modeling and reuses the infrastructure tailored to the standard AR paradigm, such as Flash Attention. However, that does not mean reAR is not applicable to other architectures. Following your suggestions, we added results on VAR, though, due to time constraints, we did not tune any hyperparameters for training or inference:
>
> | Method |FID|IS|
> |---|---|---|
> |VAR-d16|3.55| 274.4|
> |VAR-d16+reAR |3.39|276.6|
>
> These results further demonstrate the generalization capability of reAR. Our core insight is that whenever a model is (1) implemented in an autoregressive manner and (2) relies on a discrete tokenizer, a generator-tokenizer inconsistency inevitably arises, regardless of the specific architectural variant. Consequently, we believe that reAR provides an effective and generalizable solution across different forms of discrete autoregressive models. We appreciate the reviewer’s suggestion again and have added the supplementary results in the revision.
>
> > The term "context" should be defined...
>
> We adopt the definition from existing literature [3,4,5,6] where, context is considered as the historical token used for predicting the next token. We appreciate the reviewer’s kind suggestion and have revised this in our Preliminary Section.
>
> > Figure 1 should be explained in more detail. ...?
>
> The orange cat’s upper body is in a sitting posture, while its lower body is flipped with the belly facing upward, resulting in an unnatural pose (see Lines 75–76). The top and bottom images may appear similar despite having different token indices because distinct token sequences can correspond to nearby embeddings (see Lines 77–85). This observation further supports our motivation. Please refer to Section 3.1 (Figure 3b) and Appendix B for additional details.  Thank you for this suggestion; we have added these clarifications directly to the caption.
>
> **Again, we thank the reviewer for the recognition of our work! We welcome additional questions and comments.**
>
> [1] VQGAN, CVPR 2022.
>
> [2] VAR, NeurIPS 2024.
>
> [3] RAR, arXiv 2024.
>
> [4] Scheduled Sampling for Sequence Prediction with Recurrent Neural Networks, NeurIPS 2015.
>
> [5] Professor Forcing: A New Algorithm for Training Recurrent Networks, NeurIPS 2016.
>
> [6] Self-Forcing: Bridging the Train-Test Gap in Autoregressive Video Diffusion, arXiv 2025

---

> > ### Author Response · Authors · 2025-11-27
> >
> > Dear Reviewer YZ8R,
> >
> > We sincerely appreciate your thoughtful comments and the opportunity to clarify our scope and provide additional VAR experiments to demonstrate the generalization of our method. As the discussion period concludes in a few days, we would be happy to address any remaining questions or provide further details for your final evaluation.
> >
> > Thank you again for your time and thorough review.
> >
> > Best regards,
> >
> > The Authors

---

### Author Response · Authors · 2025-11-21
**Common Response**

Dear Reviewers and AC,

We sincerely appreciate your time and thoughtful feedback on our manuscript. We thank all reviewers for their constructive suggestions. In response, we have carefully revised and strengthened the paper. The manuscript now includes:

- Experiments on VAR-d16 retrained with reAR (Appendix A)

- Clock-wise training time cost overhead of reAR (Appendix A)

- Additional discussion on the choice of the layer in the main text apart from the Appendix (Section 3.2)

- Limitation and Future Work (Appendix E)

For ease of review, important revisions are highlighted in red, and minor issues are simply fixed. We hope these updates clarify our contributions and improve the overall presentation. We also invite reviewers to evaluate the provided anonymous code in the Reproducibility Section for testing  our method.

Thank you again for your time and consideration.

Sincerely,
The Authors

---

### Author Response · Authors · 2025-11-25

With the discussion period having run for approximately two weeks, we want to thank Reviewer RcEf for raising their score. We kindly invite the reviewers to let us know if we have addressed their concerns. We will provide further clarifications if needed.

---

### Author Response · Authors · 2025-12-02

**Dear Area Chairs**,

We sincerely thank you for taking on our submission and for your efforts in assessing our work. To assist your evaluation, we briefly summarize the review progress and highlight the key recognitions from the reviewers. **Before the OpenReview incident, the overall ratings improved from (6, 6, 6, 4) to (8, 6, 6, 6): Reviewer RcEf raised the score from 4 → 6 on Nov 25, and Reviewer n9Nt raised the score from 6 → 8 on Nov 26.** For the two remaining reviewers (both scoring 6), we believe our additional clarifications and extensive experiments fully addressed their concerns.

## Summary of Our Work

We identify **generator–tokenizer inconsistency** as a fundamental and previously overlooked flaw in *discrete* visual autoregressive (AR) models. Through targeted analysis, we show that this issue arises from both embedding unawareness and exposure bias. To resolve it, we introduce **reAR**, a simple and scalable regularization framework that:

- Injects *current-token* embedding information at the first layer,
- Aligns *next-token* embeddings at a deep decoding layer, and
- Uses lightweight, annealed perturbation to mitigate exposure bias.

reAR incurs **negligible overhead** and demonstrates strong generalization across different forms of visual AR models. Notably:

- **With a naive VQGAN tokenizer and a 461M model, reAR improves FID from 3.02 → 1.86, surpassing VAR (1.92, 2.0B) and MAR (1.98, 479M).**
- **With AliTok and only 177M parameters, reAR achieves FID 1.42, competitive with REPA (675M, requiring DINO-v2).**
- **Inference remains unchanged**, preserving compatibility with KV cache, Flash Attention, and existing AR frameworks such as VLLM.

## Reviewers’ Positive Feedback

- **Novelty**: A clear identification of a fundamental and previously under-explored limitation called tokenizer–generator inconsistency in visual AR models.
- **Motivation and Analysis**: Well-designed analyses that convincingly support the problem formulation.
- **Simplicity and Versatility**: A simple, lightweight, and broadly compatible strategy applicable to AR models across diverse discrete tokenizers and backbones.
- **Efficiency and Effectiveness**: Consistent and significant empirical gains across multiple architectures, achieving superior performance with smaller models compared to methods relying on more sophisticated tokenizers or generation paradigms.

## Responses to Reviewers’ Concerns & Major Revisions

During the discussion period, we added substantial experiments and clarifications:

**Generalization Across Models**
- The **original submission** already evaluates reAR on VQGAN, TiTok, and AliTok across different backbones and model scales; we further clarified these points to reviewers.
- We additionally included **VAR-d16** results without tuning, demonstrating that reAR also generalizes to non-standard decoder-only Transformers.

**Regularization Layer Analysis**
- The **original submission** includes detailed methodology and recommended practice in Section 3.2, comprehensive ablations in Section 4, and additional CKA-based analysis in Appendix C.2; we emphasized these findings in our responses to reviewers.

**Hyperparameter Sensitivity**
- The **original submission** includes ablations showing that reAR is robust to the choice of regularization layer, noise schedule, and regularization weight. The default configuration (perturbation scheme, layer position, weight) generalizes well across all tokenizers and AR models, which we clarified to reviewers.

**Training Overhead**
- We provided a quantitative breakdown showing that reAR adds only **0.3%** training cost while improving FID from 3.02 → **1.86**; inference remains unchanged.

---

We hope this summary provides a clear overview of the contribution of reAR. **Thank you again for your time and for taking on this assignment under challenging conditions.**

Best Regards,
The Authors

---

### Meta-Review · Area_Chair_X6wV · 2026-01-18

**Summary:**

Overall, reviewers agree that the proposed method is simple and broadly useful. Initial concerns were mostly about the evaluation benchmark, novelty, hyperparameters, and additional evidence for generalization. The rebuttal added experiments that addressed most concerns. The only negative reviewer also agreed to raise the score. Therefore, I recommend acceptance of this paper.

**Reviewer Concerns:**

There is only a little concern about "novelty" that remains not fully addressed by the rebuttal, but it is not detrimental.

**Reviewer Scores:**

The reviewers are likely to reach a consensus on positive reviews of this paper.

---

### Decision · Program_Chairs · 2026-01-26

Accept (Poster)